



**Strong downdrafts preceding rapid tropopause ascent and their potential to**
**identify cross-tropopause stratospheric intrusions**
Feilong Chen[1], Gang Chen[1*], Chunhua Shi[2], Yufang Tian[3], Shaodong Zhang[1],
Kaiming Huang[1]
[1]School of Electronic Information, Wuhan University, Wuhan 430072, China.
[2]Key Laboratory of Meteorological Disaster, Ministry of Education, Nanjing
University of Information Science &Technology, Nanjing 210044, China.
[3]Key Laboratory of Middle Atmosphere and Global Environment Observation,
Institute of Atmospheric Physics, Chinese Academy of Sciences, Beijing 100029,
China
*Corresponding author: Gang Chen (g.chen@whu.edu.cn)
**Abstract:**
The capability of measuring 3-dimensional wind and tropopause structure with
relatively high time and vertical resolution makes VHF radar a potentially significant
tool for studying various processes of the atmosphere. Here the potential detection of
possible stratospheric intrusion events is discussed using the Beijing MST radar located
at Xianghe (39.75°N, 116.96°E). During the passage of a cut-off low in late November
2014, a deep V-shaped tropopause structure, and strong downdrafts (>0.8 m/s)
immediately preceding the rapid tropopause ascent (>0.2 km/h) were observed. Within
the height region of the downdrafts, the 'normal' radar-tropopause layer seems to be
destroyed (weakened) with the decreased echo intensity. Analysis results from global



reanalysis and the satellite data, as well as the trajectory model have shown the clear
evidence of the downward stratospheric intrusions (dry ozone-rich and depleted
methane air) associated with the strong downdrafts. According to the previous studies
and the present case observation, the strong downdrafts preceding rapid tropopause
ascent are considered as a significant signature of stratospheric intrusions. Twenty
typical cases of such strong downdrafts, occurring during various synoptic processes in
different seasons, have been presented and 16 of them are exactly associated with some
form of stratospheric intrusions. Four years (2012-2015) of such downdrafts are further
discussed. The observations reveal that the strong downdrafts preceding the rapid
tropopause ascent can be a valuable diagnostic for monitoring intrusion events, which
will gain a better understanding of stratospheric intrusions in VHF radar observations.
**Keywords:** Stratospheric intrusions; strong downdrafts; rapid tropopause ascent; MST
radar; VHF radar; cut-off low





## 1.  Introduction

The tropopause is a stable transition zone separating the vertically stable stratified stratosphere from the active free troposphere. The stratospheric and tropospheric air are remarkably different in their chemical and dynamical characteristics. The stratosphere is dominantly high in ozone and potential vorticity (PV) content and low in water vapor (WV) and methane ($CH_4$) concentration, while the troposphere is just on the contrary (Holton et al., 1995). Consequently, the natural stable tropopause layer, characterized by strong gradients of trace constituents and wind speeds, plays an important role in stratosphere-troposphere exchange (STE) processes. In other words, the layer is a significant barrier for the atmospheric transport between stratosphere and troposphere (Mahlman, 1997). From a long-term point of view, the long-term seasonal variation of the tropopause height determines the seasonal variation of the flux of stratospheric air into the free troposphere (Appenzeller et al., 1996). Under the global climate warming (e.g. the continuing rise in $CO_2$), the tropopause variation is also a significant factor that must be considered when comes to the recovery of the stratospheric ozone (Butchart et al., 2010; Chipperfield et al., 2017). On the other hand, the short-term tropopause variability is sensitive to various meso– and small–scale atmospheric processes, during which the folding/intrusion events commonly occur. This characteristic of the tropopause change are sometimes directly used to detect the tropopause folds (e.g. Rao et al., 2008; Alexander et al., 2012, and references therein), but are less, if any, directly used to identify stratospheric intrusions. More detailed analysis of the variability of high-resolution tropopause height and of course some other



parameters (e.g. 3-dimentional wind), and how the stratospheric air transport across the
tropopause into the troposphere will help us to yield better understanding of the
downward stratospheric intrusions (e.g. Sprenger et al., 2003; Leclair de Bellevue et al.,
2007; Das et al., 2016).

Photochemical production within the troposphere, although, is the main source of

tropospheric ozone, the influence of the downward stratospheric intrusions on the
tropospheric ozone content cannot be ignored (Oltmans and Levy II, 1992; Monks,
2000; Stevenson et al., 2006). Stratospheric intrusions bring dry ozone-rich air down
into the free troposphere (e.g. Stohl et al., 2000; Sørensen and Nielsen, 2001) and
sometimes even deep to the surface (e.g. Gerasopoulos et al., 2006; Ding and Wang,
2006; Lefohn et al., 2011). By now, it is well established that these intrusions of
stratospheric origin will significantly influence other trace gases (such as hydroxyl
(OH)) in the troposphere (Holton et al., 1995). These influences then will further
contribute to the change of radiative balance (Ramaswamy et al., 1992) and play an
important role in the radiative forcing of global climate change (Holton et al., 1995). It
is true that stratospheric intrusion events occur all over the world and in any season.
However, they are highly episodic in both vertical and isentropic (horizontal) directions
(Chen, 1995). Various dynamical and physical processes have been proposed to be
responsible for extra-tropical intrusion events. These mainly include tropopause folds,
stratospheric streamers and break-up, cut-off lows (COLs), wave breaking, and
mesoscale convective activities and thunderstorms (Stohl et al., 2003).

The certain dynamical and chemical characteristics of stratospheric air allow the



tracers, such as dry ozone-rich and high PV, to be proper indicators for the intrusions
penetrating down into the troposphere. Various methods are available to detect intrusion
events based on these tracers. Among them, balloon-borne ozonesonde sounding are
without doubt one of the most appropriate tools, but is limited by coverage (He et al.,
2011) and not possible to obtain continuous profiles with fine temporal resolution. In
contrast, the satellite-borne remote sensing instruments, such as Atmospheric Infrared
Sounder (AIRS), can provide nearly global coverage of various trace gases but have
limitations in vertical and temporal resolution. Another method for studying transport
processes is trajectory model, from which the backward trajectories can provide
valuable information on the possible sources of the trace gases (e.g. Elbern et al., 1997).

By far, large-scale STE has been widely studied and is fairly well understood, but

the details of small scale intrusions are still remain uncertain (Holton et al., 1995).
Kumar and Uma (2009) reported that the dearth of direct vertical wind measurements
in the vicinity of the tropopause may be responsible for the lack of detailed fine
observations of smaller scale intrusions.

Very-High-Frequency (VHF) radars, comparing the tools mentioned above, are

capable to provide 3-dimensional wind and tropopause height continuously with both
high temporal and spatial resolution and can operate unmanned continuously for 24
hours per day under any weather conditions. During the past two decades, VHF radar
measurements were commonly used to assist to study the stratospheric intrusions.
However, it still remains uncertain in many aspects when using only the VHF radar to
identify intrusion events. Complicated and changeable atmospheric processes make it





difficult to identify the intrusion events by only radar data. The research by Hocking et
al., (2007) have achieved a development in this issue. They found that the rapid ascent
in radar-derived tropopause altitude (>0.2 km/h) can be a valuable diagnostic for
possible stratospheric intrusions. However, it does not always work (e.g. He et al., 2011)
and remains uncertain when purely using the information of radar-determined
tropopause.
The central objective of the present study is to discuss the signature of downward
cross-tropopause intrusions using both the measurements of tropopause height and
vertical wind by the Beijing MST radar. This study is carried out mainly via a detailed
case observation during the passage of a COL and other general cases associated with
various atmospheric processes. Our discussion mainly focused on the potential of the
MST radar data to identify possible intrusion events, which is the main point of this
paper. In section 2 the datasets used in this paper are described, section 3 presents
detailed results and discussion, and section 4 gives the conclusions.





**2. Dataset**
2.1. MST radar data and tropopause detection
The Beijing MST radar located at Xianghe, China (39.75° N, 116.96° E, 22 m
above sea level) is a VHF radar operated at 50 MHz and installed in 2010 based on the
first phase of Chinese Meridian Space Weather Monitoring Project (Chinese Meridian
Project for short) (Wang, 2010). The radar antenna array consists of 24×24 three-
element Yagi to produce an average power aperture product of $3.2 \times 10^8$ Wm² and
maximum directive gain of 34.8 dB. It operates radiation pattern with 172 kW peak
power and 3.2° half-power beam width. More detailed information of the radar system
can be found in Chen et al. (2016). Routine low mode data were used for present study
with 0.5 h time resolution and 1 μs coded pulse, which provides 150 m vertical
resolution. Details of the low mode setup used in this study are given in Table 1.
It has long been known that VHF radar reflectivity is proportional to the mean
generalized refractive index gradient M, which is a function of humidity variation and
static stability and given by (Ottersten, 1969) as follows
$M = -77.6 \times 10^{-6}(p/T)(dln\theta/dz)$

$\cdot \{1 + 15500q/T[1 - (dlnq/dz)/(2dln\theta/dz)]\}$          (1)

where $p$ is the atmospheric pressure (hPa) $T$ is the temperature (K), $\theta$ is the potential
temperature (K) and $q$ is the specific humidity (gg⁻¹). According to the second and third
terms of the equation (1): large humidity variation contributes to the echo from the
lower and middle troposphere. From the first term: the radar backscatter power is
proportional to the static stability, which in fact is directly proportional to the potential



temperature gradient. The tropopause, near which a strong potential temperature
gradient exists, will lead to strong radar echoes in vertical incidence, as well as large
radar aspect sensitivity (as shown in Figure 1). Radiosonde data used in this paper were
received from the GTS1 type digital radiosonde launched from Beijing Meteorological
Observatory (39.93 °N,116.28 °E, station number 54511), which is less than 45 km to
the MST radar site. The black line in Fig.1 denotes the lapse-rate tropopause (LRT)
defined using the temperature lapse rate (World Meteorological Organization (WMO),
1986). Applying the characteristic (partial specular reflection) mentioned above, the
tropopause can be detected and its height determined by VHF radars (Gage and Green,
1979). It has received widespread application around the world, either in middle
latitudes (e.g. Hocking et al., 2007), polar regions (e.g. Alexander et al., 2012), and
tropical regions (e.g. Yamamoto et al., 2003; Das et al., 2008). Here, the radar-
determined tropopause (RT) height is defined as the lower edge (the height above 500
hPa with largest power gradient) of the secondary maximum backscattered echo power
(shown in Figure 1a as the orange circle). This definition of RT is similar to that in the
studies of Alexander et al., 2012 and Ravindrababu et al., 2014.

In the present study, the MST radar mainly provides continuous measurements of

backscattered echo power, 3-D wind, and RT height with time resolution of 0.5 hour. In
addition, the radar aspect sensitivity that expressed as the ratio between vertical ($p_v$)
and oblique ($p_o$) beam echo power is mainly caused by the horizontally stratified
anisotropic stable air and thus will be used as potential signature of stratospheric
intrusions in the troposphere (e.g. Kim et al., 2001). The backscattered echo power



given here is expressed as relative power in decibels (dB). In order to reduce the random
noise, the profile of $p_v$ is smoothed by a 3-point running mean in altitude. Note that
the data that are heavily contaminated will be eliminated from our datasets. The data of
Dec. 2015 and Sep. 2015 are excluded.
2.2. AIRS satellite data

The AIRS instrument on NASA Aqua/EOS polar orbit satellite is a 2378 channel

nadir cross-track scanning infrared spectrometer. It can provide profiles of a number of
trace gases, including ozone and $CH_4$ (Susskind et al., 2003). The footprint of these
retrieval data is of 45 km by 45 km and their most sensitive region is in an altitude range
of 300-600 hPa. Many studies have shown that these AIRS retrieval constituents are
useful indicators for detecting stratospheric intrusions. He et al. [2011] suggested that
AIRS can observe the enhanced tropospheric ozone that is of stratospheric origin.
Xiong et al. [2013] reported that AIRS is capable of observing abnormal depletion in
$CH_4$ in the troposphere during intrusions. AIRS offers good latitude-longitude coverage.
Here we use version 6 of the AIRS Level-3 ozone and methane retrieval products.
2.3. Meteorological reanalysis

European Centre for Medium-Range Weather Forecasts (ECMWF) reanalysis

ERA-interim data are also used. After Nov. 2000 the data are based on the T511L60
version available with a 6-h temporal resolution and $3^o \times 3^o - 0.125^o \times 0.125^o$
latitude-longitude grid. The dataset from 15 isentropic and 37 pressure levels with
$0.5^o \times 0.5^o$ grid are applied for present study.
2.4. HYSPLIT model



Backward (forward) trajectories in given starting locations are capable to
reproduce the sources (destinations) of the air parcel that will allow us to examine the
intrusions of stratospheric origin in the troposphere (e.g. Elbern et al., 1997). The
Hybrid Single Particle Lagrangian Integrated Trajectory model (HYSPLIT) developed
by the National Oceanic and Atmospheric Administration (NOAA)'s Air Resource
Laboratory (ARL) (Rolph, 2003; Stein et al., 2016) is applied to calculate the backward
and forward trajectories. The calculation method of the model is a hybrid between the
Lagrangian approach and the Eulerian methodology. In this paper, Global Data
Assimilation System (GDAS) datasets are adopted for driving the HYSPLIT.



## 3.    Results and discussion

3.1. Meteorological synoptic situation

On the morning of 29 November 2014, a 500-hPa trough developed on the western side of Lake Baikal (Western Siberia). The trough moved southeastward and extended equatorward and its bottom separated from the westerlies in the afternoon of 30 November 2014 (Fig. 2b), forming a COL near the radar site. The black stars in Figure 1 and other figures indicate the location of the radar site. On the following days, the COL system moved northeastward gradually (Fig. 2b) and finally stayed over eastern Russia near Sakhalin Island until it reconnected and merged to the westerly flow. 315 K isentropic PV patterns have shown the coarse resolution features of intrusions from the polar reservoir across the tropopause into the midlatitude troposphere. The PV streamer curved and rolled up cyclonically along the western flank of the COL (Fig.2b).

Fig. 3 shows the time series of hourly surface meteorological parameters over the Beijing station. The data are obtained from the Chinese National Meteorology Information Center and is less than 50 km from the MST radar site. As the dry-cold air invasion accompanied with the COL travelled deeply into the planetary boundary layer, it brought severe weather to the surface, including a rapid decrease in temperature and humidity, and rapid increase in surface wind and sea level pressure. The humidity decreased from ~85 to 12 percent within less than 8 hours. It is well established that the polar-type COLs have strong potential to trigger deep convection (Price and Vaughan, 1993). To examine the potential convection, maps of high quality Climate Data Record (CDR) of daily Outgoing Longwave Radiation (OLR) are displayed in Fig. 4. During

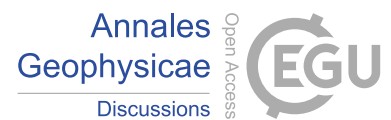

the development of the COL, a local region with abnormal low OLR value was clearly
observed near the radar site on 29 Nov. (Fig. 4b). The Satellite-observed cloud top
temperature also showed the low values corresponding to the low OLR (figure not
shown), indicating a convection may be generated near radar side on 29 Nov.. Please
note that we didn't observe such low value either in OLR (Fig.4c, d) or in cloud top
temperature near the radar side on 30 Nov. and 1 Dec.. The time for all the observations
in this paper is showed in Universal Time (UTC) which is eight hours behind Beijing
standard time (LT=UT+8).
**3.2. MST radar observations**

Radar echo power, horizontal wind vector, vertical wind, and radar aspect

sensitivity are plotted in Figure 5 as function of height and time during the passage of
the COL. Time variation of RT (black line) and LRT (black crosses) heights are also
displayed. The RT height first experienced a rapid descent, and then increased rapidly,
forming a deep V-shaped structure of ~4 km depth. The vertical velocity of the RT
height variation (both the rapid descent and ascent branches) reaches up to 0.28 km/h.
The rapid RT variation in altitude is in fact the response of the tropopause fold below
the jet stream, which will be well represented in Fig. 9a. Rapid variation in RT height
remained a region with low echo power (marked by R on Fig. 5a) and low aspect
sensitivity (marked by R' on Fig. 5d) where they should be normally high value within
the 'normal' tropopause layer. Unlike the RT height, the radiosonde LRT altitudes are
nearly constant during the COL passage. In normal conditions, RT agrees well with the
LRT altitude, such as indicated by Fig. 6a. However, large differences, of order of 2.5





km (as shown in Fig.6b at 12 UT 30 Nov.), are observed between LRT and RT in
altitude during the passage of the COL as expected. It is the difference in definition that
contribute most to the large differences, especially under the tropopause fold conditions.
It is worth noting that, in Fig.6b, although there is no clear reversion in the radiosonde
temperature profile within the height of RT, the RT height exactly corresponds well to
the reversion of zonal and meridional wind and potential temperature gradient. Such
differences between RT and LRT heights can commonly be observed, especially during
extreme synoptic situations such as cyclone (e.g. Alexander et al., 2012).
The most important observation in this detailed case experiment is the strong
downdrafts (hereinafter inferred to as main downdrafts) observed immediately
preceding the rapid RT ascent (Fig.5c). The radar echo power sharply weakened (dotted
rectangle in Fig.5a) and the wind direction changed rapidly (Fig.5b, change from
dominant southerly wind to dominant northerly jet) within the height region of the main
downdrafts. As mentioned previously, abnormal low value in OLR and cloud top
temperature indicates the possible occurrence of convective activity on 29 Nov., but
nothing special appeared on 30 Nov. near radar site. Consequently, we preliminarily
consider that the main downdrafts occurred near 07 UT 30 November might not be
produced directly by convective activity. Here, the accurate origin of the main
downdrafts will not be discussed in detail, and it is also beyond the scope of present
study.
The research by Hocking et al. (2007) has suggested that the rapid RT ascent (>0.2
km h$^{-1}$) can be a valuable indicator for the occurrence of stratospheric intrusions. Here



in this paper, the main downdrafts preceding the rapid RT ascent observed by the
Beijing MST radar are thus suspected to be an important feature or response of some
form of vertical stratospheric intrusions. It is indeed reasonable. Firstly, as the
tropopause descends (folded downward), it will displace stratospheric air into the
troposphere (e.g. Hoskins et al., 1985). Secondly, the main downdrafts will act as an
effective way to weaken the tropopause by means of continuously impinges on the
tropopause, through which the stratospheric air is permitted to penetrate down into the
free troposphere (e.g. Hirschberg and Fritsch, 1993; Kumar, 2006). In addition, after
the main downdrafts, the observed region near the upper troposphere with strong
backscatter echoes (marked by Q) and especially with abnormal high aspect sensitivity
(marked by Q') may also be a weak signature of the possible intrusions. In normal
conditions in the upper-troposphere, they are usually low in value (such as the region
marked by P and P'). As we mentioned before, the large value in radar aspect sensitivity
is mainly caused by reflection from stable atmospheric layer, such as the tropopause or
lower-stratosphere. When stable stratospheric air intrudes into the troposphere and
without mixing with the surrounding air mass, the intrusions in the free troposphere
will be reflected as abnormal large aspect sensitivity. Further direct evidence of the
relevant intrusions in dynamical and chemical aspects will be demonstrated in next
section, using satellite AIRS and global reanalysis data.

Someone may be interested to notice the laminar periodic downdrafts and updrafts

near RT height for ~16 UT 30 Nov.-2 UT 1 Dec.. It is likely to be associated with
mountain wave activity induced by northerly jet. As shown in Supplementary figure S1,



there is relatively higher topography (~1000 m mountains) located in the north of the
radar site. To examine this aspect in detail, Fig. 7b shows the wavelet spectra of the
vertical wind in the lower stratosphere (at ~12.4 km, Fig. 7a). Results reveal the wave
activity with a period of ~3.5 hours and amplitude of ~1.1 m s$^{-1}$.
3.3. Associated stratospheric intrusions
Due to the sensitivity of the AIRS retrieved ozone and $CH_4$ is between 300-600
hPa. Fig. 8 shows the 500 hPa distribution of AIRS observed ozone and $CH_4$, along
with the AIRS tropopause contour (defined based on the temperature lapse-rate). The
ozone distribution maps (left panels of Fig. 8) clearly show a large area with enhanced
tropospheric ozone (>80 ppbv) near the radar site during the passage of the COL.
Moreover, severe $CH_4$ depletion (<1840 ppbv) was also observed (right panels in Fig.8).
These features of the ozone enhancement, $CH_4$ depletion, and the corresponding low
tropopause altitude clearly support the evidence of vertical downward cross-tropopause
stratospheric intrusions on 30 Nov..
The vertical cross-section of ECMWF PV and specific humidity at 1800 UT 30
November 2014 and the daily AIRS ozone on 30 November 2014, along with a constant
latitude 40° N, is shown in Fig. 9. The corresponding vertical structure of the
stratospheric intrusions (dry ozone-rich and high PV along with low tropopause) over
regions near radar side is clearly seen. The specific humidity tracer displays less distinct
structure as compared with the other two tracers (similar as that shown by Vérèmes et
al., 2016). The cross-section of PV in Fig. 9a have demonstrated relatively finer-scale
structure of the stratospheric PV intrusions (below the jet stream), which penetrated


down deeply into ~650 hPa (~3.6 km).

3.4. Trajectory model analysis

Figure 10 shows 30h backward trajectories ending at the radar site at 18 UT 29

November (left panel) and at 18 UT 30 November (right panel). As expected, the air
masses parcel transported eastward horizontally before the occurrence of main
downdrafts (fig.10a). Whereas after the downdrafts, the trajectories clearly show that
the tropospheric air masses over the radar site are of stratospheric origin from the
western side of Lake Baikal. Trajectory results further support the evidence of
stratospheric intrusions that closely related with the main downdrafts.

On the other hand, 30-h forward trajectories starting at 00 UT 30 November (left

panel) and 00 UT 1 December (right panel) are shown in Fig. 11. It is interesting to note
that, from Fig.11a before the passage of COL, the air parcels at 4 km transport rapidly
upward (by more than 4 km within ~23 h) and northeastward to the upper-troposphere
of East Siberian. This upward and poleward transportation is associated with a warm
conveyor belt (dominate southerly flows) that is located ahead of the COL. It
contributes to transporting the tropospheric moist and polluted air (such as aerosol) into
the upper-troposphere and even the lower stratosphere (e.g. Stohl et al., 2003; Sandhya
et al., 2015). After the downdrafts, forward trajectories in fig.11b demonstrate that the
dry intrusion air parcels continue to be transported downward and southeastward to the
boundary layer or even the surface.
3.5. Strong downdrafts preceding rapid tropopause ascent and discussion





Figure 12a shows another 20 typical cases with strong downdrafts preceding rapid

RT ascent for the period Mar. 2012 and Jan. 2015 (shown placed end-to-end), and the

LRT height (plotted in crosses) and the vertical velocity of the RT (plotted in orange

line) is also plotted. These cases (marked by black rectangular boxes) are identified

based on the following criteria: 1) the amplitude of the RT ascent should exceed 0.6 km

(four range gates), 2) vertical velocities of the RT ascent excess 0.1 km/h, 3) the

downdrafts occurred preceding the RT ascent should reach at least 0.5 m/s, and the

height region of the downdrafts should pass through the RT layer. The criteria are put

forward mainly to avoid the influence of the RT spikes. Figure 12b shows the selected

9 cases of possible intrusions by means of the backward trajectories. Results show clear

evidence of possible stratospheric intrusions corresponding to the associated strong

downdrafts. Their sources are mainly from West Siberia (western side of Lake Baikal),

except for the case Tr5. Moreover, according to AIRS daily 500 hPa ozone distribution,

almost every case in Figure 12a (except for the cases labeled as A, B, C and D in Figure

12a) were associated with some form of significant ozone enhancement, indicating

intrusions of stratospheric origin (as shown in Supplementary figure S2). It is important

to note that the RT excursion velocity of all the cases is not all above 0.2 km/h and some

are lower than this value (e.g. case on 2 May. 2014). However, some form of

stratospheric intrusions were exactly observed in such case from both the trajectory and

satellite results. Therefore, the threshold of vertical velocity of the RT is set at 0.1 km/h,

rather than 0.2 km/h (Hocking et al., 2007). Large differences between RT and LRT are

also interesting to be noted on some occasions when the RT changes rapidly (such as

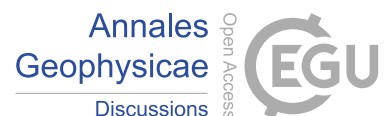

the occasion near 14 Mar. 2012).

According to the meteorological chart, the synoptic situation of those cases

identified in Fig.12a are introduced. The cases occurred on 6 Mar. 2012, 13 Jun. 2012,
and 31 Dec. 2012 seem to have a close relationship with the COL development; cases
on 13 Mar. 2012, 5 Apr. 2012. 6 Apr. 2012, 4 Jan. 2014, and 2 May 2014, seem
associated with low or high pressure systems. The remaining cases seem not associated
with any significant synoptic development. However, in terms of the distribution of
isentropic PV (generally at 315K in winter and 330K in summer), we found that the
remaining cases occurred on 3 Aug. 2013, 3 Jan. 2014, and 3 Jan. 2015 appear to be
associated with some form of stratospheric streamers and their break-up within the
previous 48h. Some cases (e.g. A and B) that appear close on the same day were
probably caused by the same system and not possible to examine the associated possible
intrusions separately using either the reanalysis or satellite data.

In the light of present understanding, the strong downdrafts preceding the rapid

RT ascent can serve as an important predictor for intrusion events, during any synoptic
processes in any season. This characteristic will be of great use and play an important
role in routine identification or prediction of stratospheric intrusions. Considering the
duration of such downdrafts, a higher time resolution of radar observations will be more
helpful. Present study have shown the duration of the majority downdrafts is generally
within 1.5-3 hours. We consider, therefore, that the radar resolution should be best
within 1h.

Although Hocking et al. (2007) have reported that the rapid tropopause ascent



(>0.2 km/h) alone can be a useful diagnostic for potential intrusion events. However,
using only the information of RT heights might lead to non-negligible errors, as
mentioned above in introduction and according to the observations in Fig. 12.
Especially on occasions when the RT ascent is between 0.1-0.2 km/h but the
corresponding true intrusions were observed, all such intrusion events will be neglected
(maybe ~2 per month, refer to Fig. 13a). Whereas on some occasions when the RT
ascent excess 0.2 km/h, but without observing true intrusion events (e.g. He et al., 2011),
these events will be misdiagnosed (maybe ~13 per month, refer to Fig. 13b). In this
sense, using the unique MST radar observations of both the RT height variability and
the vertical wind as complementary signature for identifying possible intrusion events
is very meaningful.
Figure 13 shows four years (2012-2015) of the events with rapid RT ascent (gray
bands), and the events with strong downdrafts just preceding the rapid RT ascent (black
bands). The identification criteria of such strong downdrafts are similar to that
mentioned above and the events are classified according to different value of vertical
velocity of the ascent. Among all the events with ascent velocity between 0.1-0.2 km/
h, about one-quarter (approximate 2 per month, Fig. 13a) were observed with strong
downdrafts preceding them. Whereas, as for the events with the ascent velocity >0.2
km/h, the proportion is about a half (approximate 10 per month, Fig. 13b). Here,
according to the results above, the occurrence of the strong downdrafts just preceding
the rapid RT ascent (black bands in Fig. 13) to a large degree represents the occurrence
of possible intrusions. In this way, Fig. 13 indicates that the occurrence of possible



intrusions exhibit distinct seasonal variations, with a maximum in winter and spring
minimum in summer. This is because the meso- and small-scale atmospheric processes,
such as cold air outbreaks, thunderstorms, and convective activities, are more active in
winter and spring. They are important sources for downward stratospheric intrusions.



## 4. Conclusions


Detailed case analysis of the cross-tropopause stratospheric intrusions was carried
out during a COL. Global reanalysis, satellite data, and HYSPLIT trajectories all
showed consistent evidences of dry ozone-rich, high PV, and depleted CH4 air that have
penetrated downward into the free troposphere. The key signature of the stratospheric
intrusions in the Beijing MST radar observations is the strong downdrafts just preceding
rapid RT ascent. The radar echo power decreased rapidly within the region of strong
downdrafts, after which abnormal high aspect sensitivity was recorded in troposphere.
Such high aspect sensitivity is served as another potential clue for the intrusions of
stratospheric origin. By means of wavelet spectra analysis, the periodic perturbation
with period of ~3.5 h is found in vertical wind in the lower stratosphere. This
perturbation is probably related to the mountain wave activity induced by northerly jet.
Based on the criteria mentioned in section 3.5, other 20 typical cases of strong
downdrafts preceding the rapid RT ascent between Mar. 2012 and Jan. 2015 were
presented. These events occurred during different synoptic processes in different
seasons. What counts is, almost all the cases (16 of them) are associated with some
form of intrusions observed by combination of AIRS retrieved ozone and the HYSPLIT
trajectory model. Our results show that the radar derived tropopause height and vertical
winds are strong complementary indicators to be used to infer the occurrence of the
intrusions of stratospheric origin. This will be of great use and play an important role
for the routine identification or prediction of intrusion events. However, the actual
origin of the observed downdrafts preceding the rapid RT ascent is not addressed in this





420 paper. Further combination observational experiments need to be conducted, especially

421 combined using ozonesonde soundings, to quantitative analyze the effectiveness of

422 present identification criteria for possible intrusions.





**Acknowledgment**
The authors really appreciate Prof Shira Raveh-Rubin for reading and checking the
manuscript, using the criterion in Raveh-Rubin, 2017. This work is funded by National
Natural Science Foundation of China (NSFC grants No. 41722404 and 41474132). The
authors would like to thanks the technical and scientific staff of Chinese Meridian Space
Weather Monitoring Project (CMSWMP) for their support in conducting the
experiment. The authors sincerely acknowledge the ECMWF, NASA, and NOAA Air
Resources Laboratory (ARL) for providing global reanalysis, satellite trace gases, and
HYSPLIT transport model, respectively. The MST radar data for this paper are
available at Data Centre for Meridian Space Weather Monitoring Project
(http://159.226.22.74/). The radiosonde data is available from
http://weather.uwyo.edu/upperair/sounding.html.





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



**Table**

| Radar parameter | Value |
| --- | --- |
| Transmitted frequency | 50 MHz |
| Antenna array | 24 × 24 3-element Yagi |
| Antenna gain | 33 dB |
| Transmitter peak power | 172.8 kW |
| Code | 16-bit complementary |
| No. coherent integrations | 128 |
| No. FFT points | 256 |
| No. spectral average | 10 |
| Pulse repetition period | 160 μs |
| Half power beam width | $3.2^o$ |
| Pulse length | 1 μs |
| Range resolution | 150 m |
| Temporal resolution | 30 min |
| Off-zenith angle | $15^o$ |

**Table 1.** Operating parameters in low-mode of the Beijing MST radar.





**Figures**

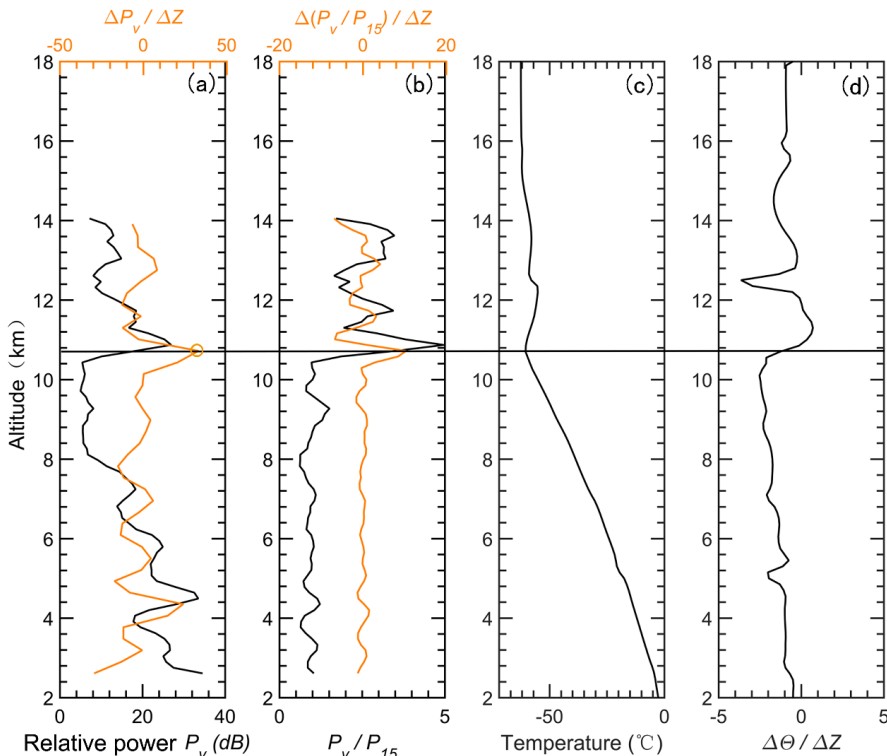


**Figure 1.** Example of the vertical height profiles of (a) the relative radar echo power
(black line, smoothed by a 3-point running mean) along with its gradient variation
(orange line), (b) the aspect sensitivity (black line, expressed as the ratio between the
vertical echo power and oblique echo power) along with its gradient variation (orange
line), observed on 12 UT 29 November 2014. The vertical profiles of simultaneous
radiosonde observed temperature and potential temperature gradient are shown in plots
(c) and (d). The black horizontal line denotes the LRT height derived from the
radiosonde temperature profile. The orange circle indicates the RT height derived from
the profile of the radar backscattered echo power.





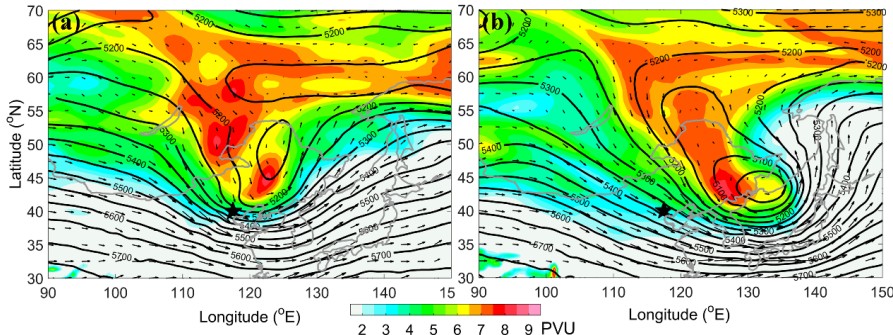

**Figure 2.** ECMWF derived isentropic PV map on 315 K surface (shaded above 2 pvu,

1 PVU=$10^{-6}$ $m^2$ K $kg^{-1}$ $s^{-1}$) and geopotential height (contoured every 50 m in solid line)

along with the wind vector (arrow) at 500 hPa (~5.5 km a.s.l.) on (a) 18 UTC 30

November 2014, (b) 12 UTC 1 December 2014. The black star shows the location of

Xianghe.



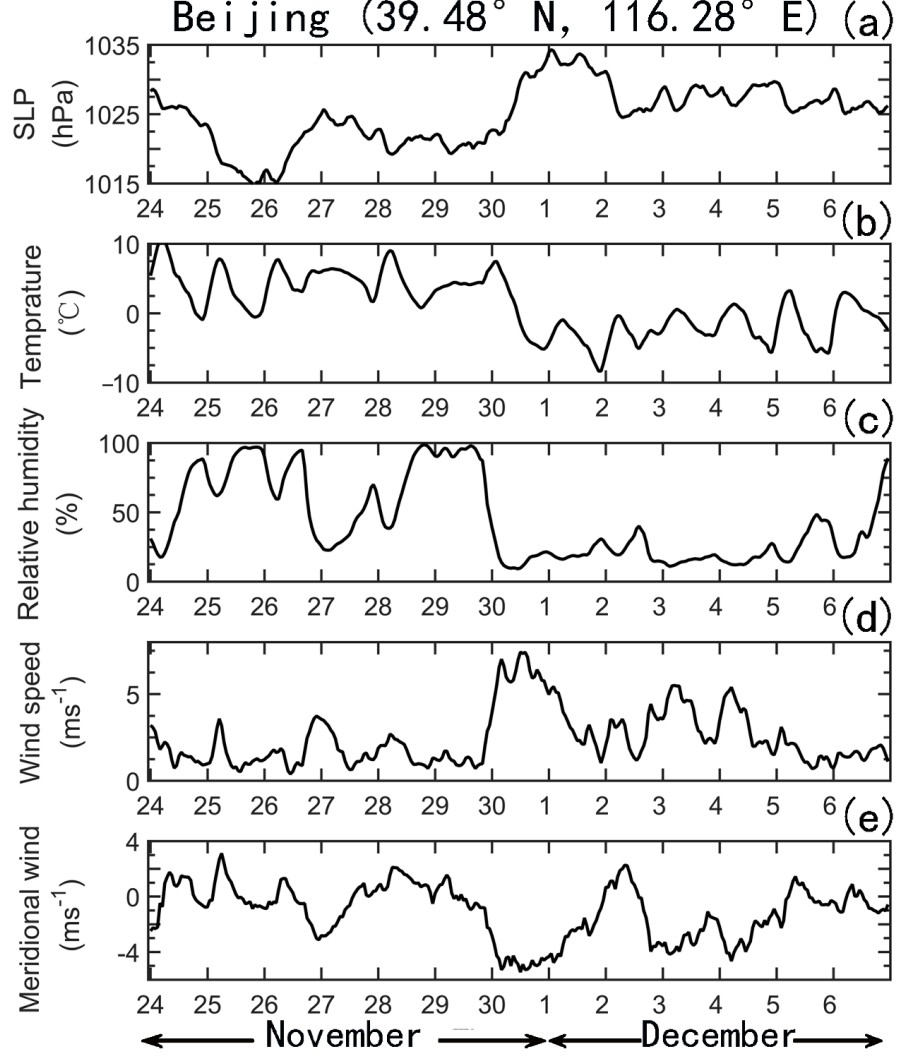

622

**Figure 3.** Time series of surface (~1.2 m above the surface) hourly meteorological

measurements of (a) sea level pressure, (b) temperature, (c) relative humidity, (d)

horizontal wind, and (e) meridional wind during the period 24 Nov.-6 Dec. 2014,

observed over the Beijing station (39.4° N, 116.2° E, 31.3 m above sea level).

627



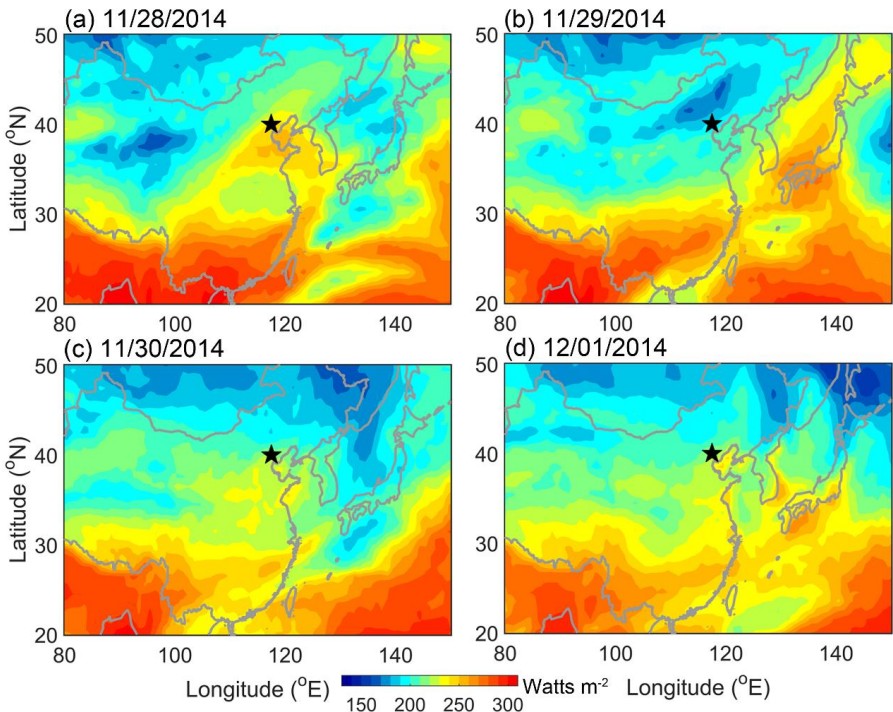

628

**Figure 4.** Contour maps of the high quality Climate Data Record (CDR) of the daily

Outgoing Longwave Radiation (OLR), derived from the NOAA high-resolution

infrared radiation sounder (HIRS) on (a) 28 Nov., (b) 29 Nov., (c) 30 Nov., and (d) 1

Dec. 2014. The black star shows the location of Xianghe.

633

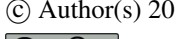



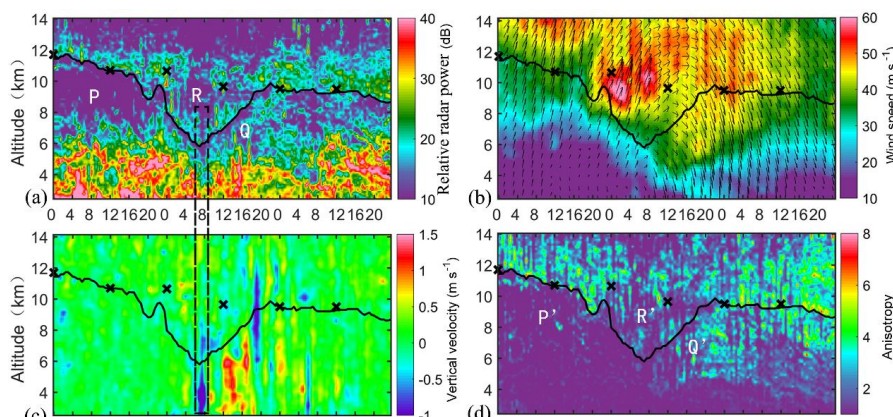

**Figure 5.** Altitude-time section of (a) the radar backscattered echo power in zenith direction, (b) the horizontal wind speed along with wind vector, of which the up and down arrows represent north and south respectively, and left-right is west-east, (c) the vertical velocity, and (d) the aspect sensitivity, observed by the Beijing MST radar from 29 November to 1 December 2014. The black curve shows the radar-determined tropopause, as defined in section 2.1. The dotted rectangle highlights the strong downdrafts immediately preceding the rapid tropopause ascent. The positions of the LRT tropopause heights, derived from the nearly simultaneous collocated GPS radiosonde temperature profile, are marked by crosses.




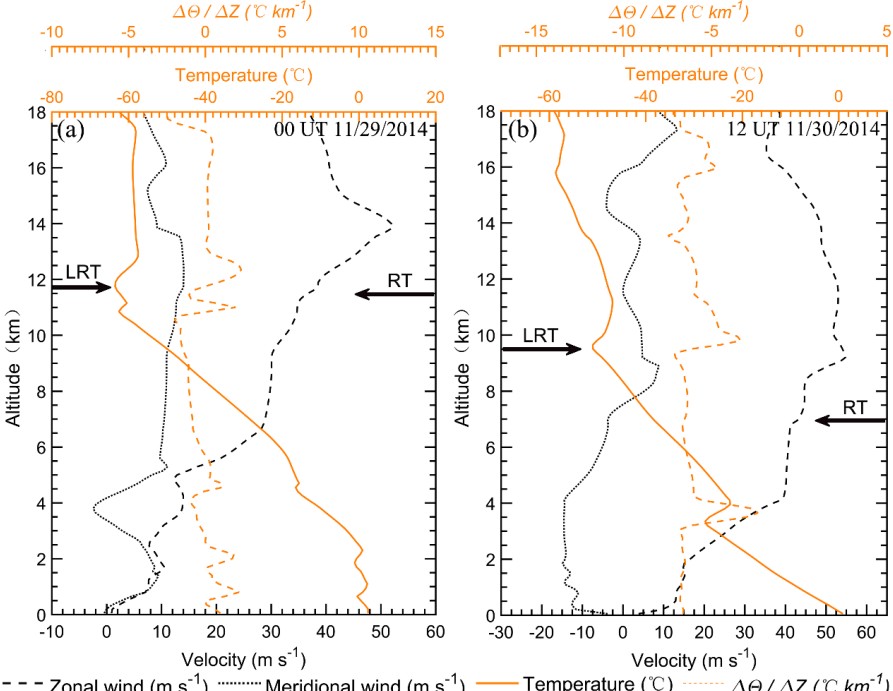

- - - Zonal wind (m s⁻¹)  ......... Meridional wind (m s⁻¹)  —— Temperature (℃)  ------- ΔΘ / ΔZ (℃ km⁻¹)
**Figure 6.** Vertical profiles of zonal wind, meridional wind, temperature, and potential
temperature gradient derived from the GPS radiosonde measurements, at (a) 0000 UTC
29 November 2014 and (b) 1200 UTC 30 November 2014. The bold arrows on the left
and right side of each panel indicate the radiosonde derived LRT tropopause and radar-
derived tropopause height, respectively.



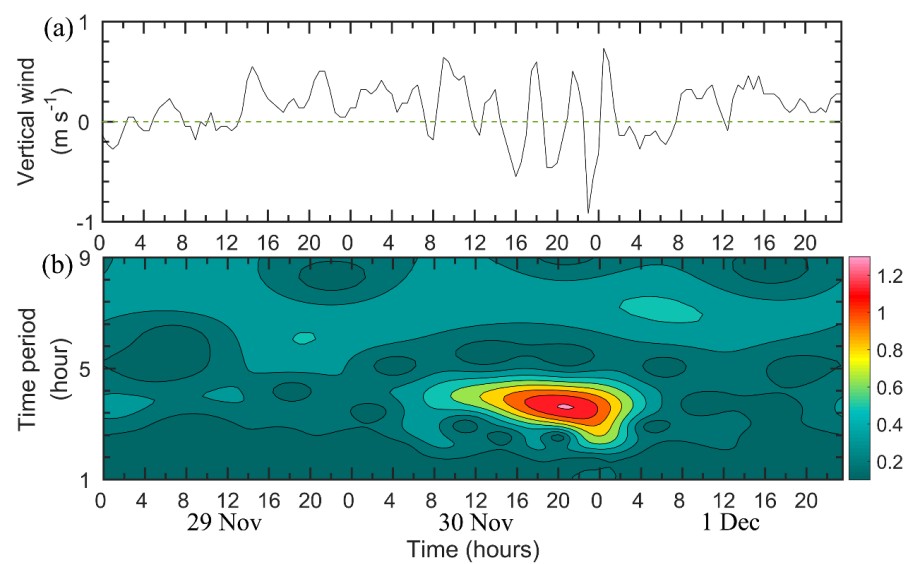


**Figure 7.** (a) Radar derived vertical velocity variations with time and (b) wavelet

spectra analysis of the vertical velocity at ~12.4 km in the lower stratosphere.





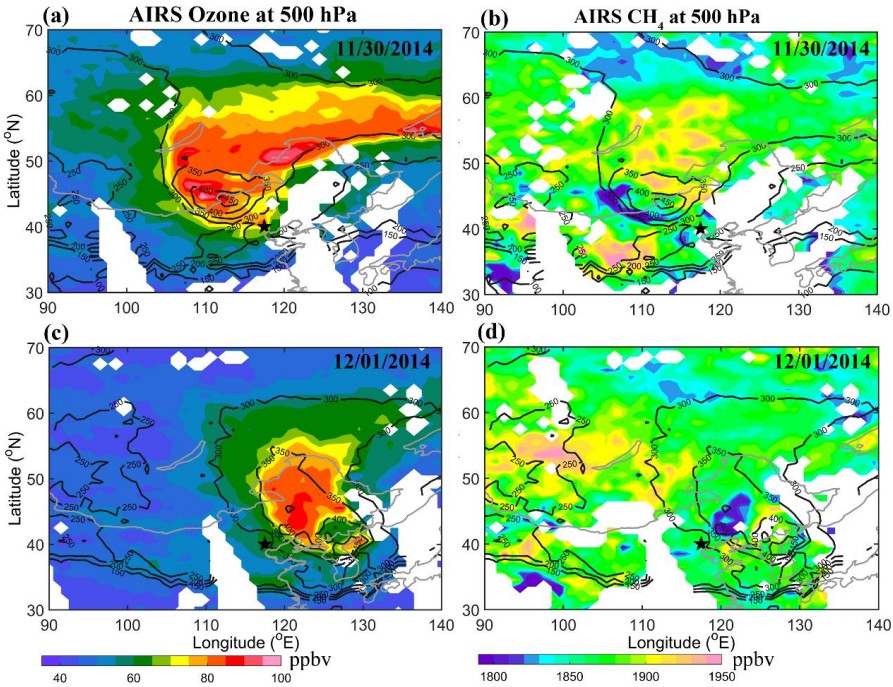

**Figure 8.** 500 hPa Ozone (left panels) and methane CH₄ (right panels) distribution
along with the tropopause height contour, derived from the AIRS satellite observations.
The top and bottom plots show the data of 30 Nov. 2014 and 1 Dec. 2014, respectively.
The black star indicates the location of Xianghe.





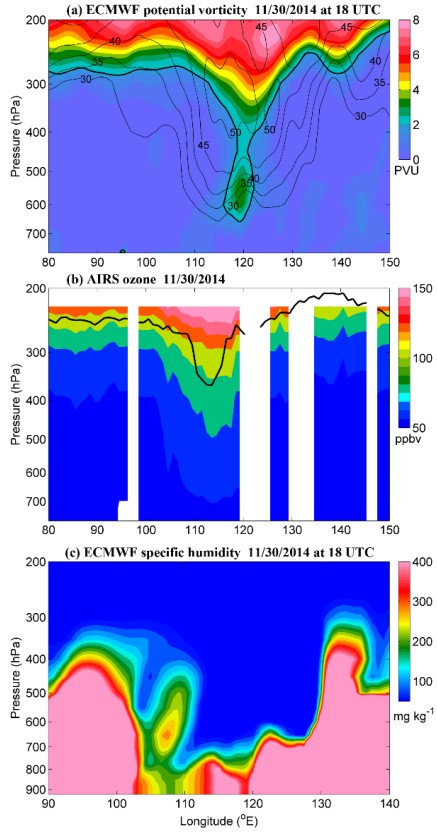


**Figure 9.** Longitude-pressure cross section of (a) ECMWF PV (colors, in pvu) along

with horizontal wind contour (thin black line, m/s) at 18 UTC on 30 Nov. 2014, (b)

AIRS ozone mixing ratio (colors, in ppbv) along with tropopause height (black line) on

30 Nov. 2014, and (c) ECMWF specific humidity (colors, in mg kg$^{-1}$) at 18 UTC on 30

Nov. 2014, at a constant latitude 40° N (nearest grid point in the latitude of Xianghe).

The bold line in (a) marks the isotropic line of PV at 2 pvu.





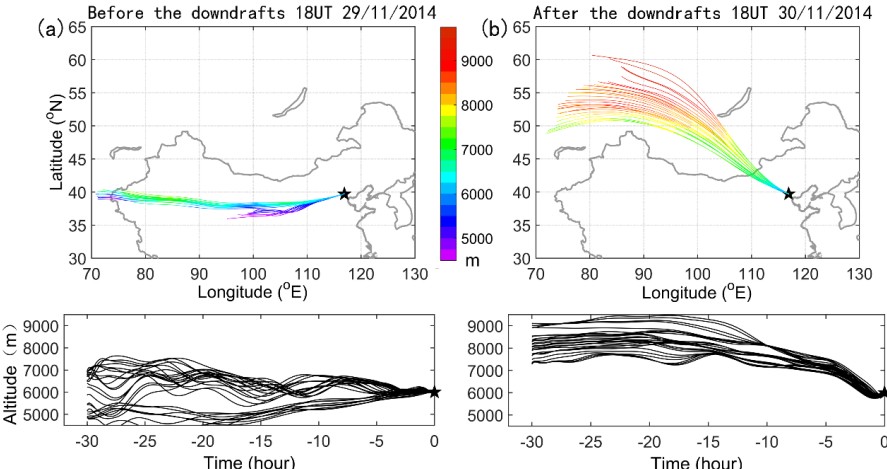

668

**Figure 10.** Illustration of 30 h three-dimensional backward trajectories ending at

Xianghe using National Oceanic Atmospheric Administration (NOAA) HYSPLIT

model: (a) before the main downdrafts at 18 UT on 29 November 2014, and (b) after

the main downdrafts at 18 UT on 30 November 2014. The HYSPLIT ensemble consists

of 27 trajectories. Upper plots show the horizontal projection of the trajectories, and the

lower plots show the corresponding time-height vertical displacement of the trajectories.

675





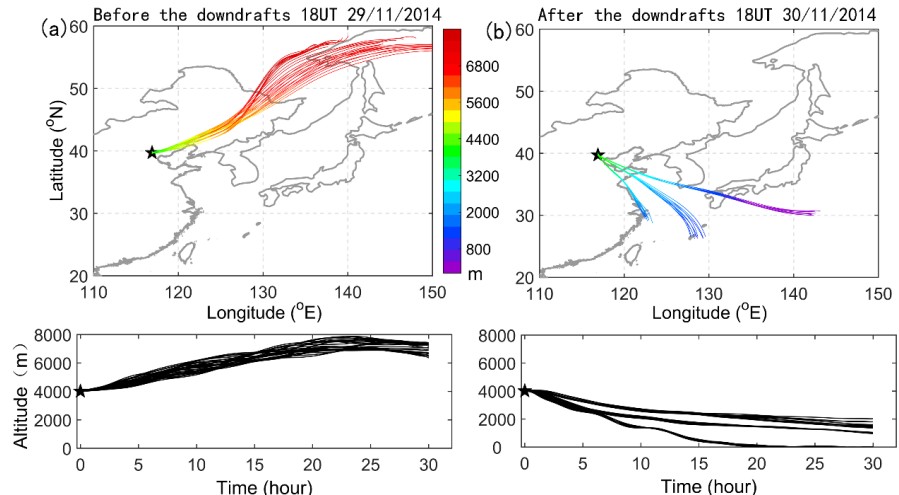

**Figure 11.** Same as Fig.10 but for three-dimensional forward trajectories starting at

Xianghe: (a) before the main downdrafts at 00 UT on 30 November 2014, and (b) after

the main downdrafts at 00 UT on 1 December 2014.

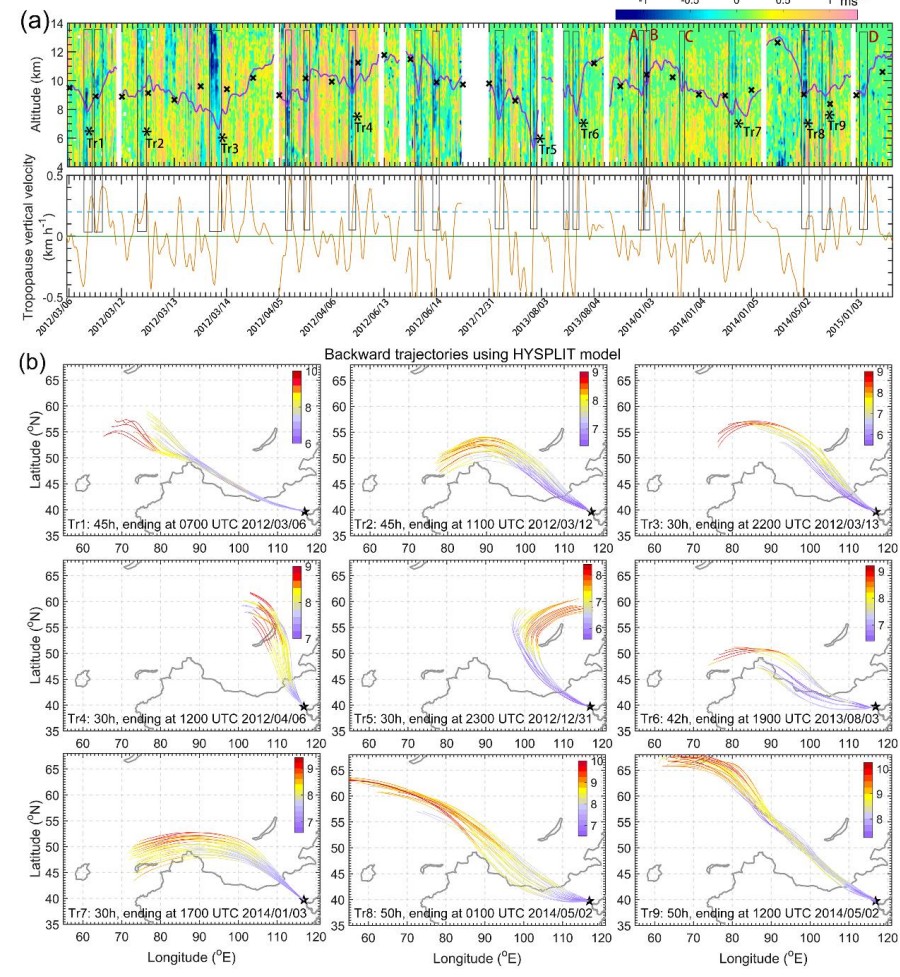

**Figure 12.** (a) Height-time section of several episodic observations of the radar-derived

vertical wind (colors in m/s) along with RT height (purple bold line) and LRT height

(bold crosses), between Mar. 2012 and Jan. 2015. The corresponding vertical velocity

of the RT (orange line) is plotted in the lower panel of (a), dotted blue line indicates the

value of 0.2 km/h. Dates for the observations are displayed as year/month/day. Black

rectangular boxes represent the strong downdraughts (absolute value $\geq 0.5$ m/s)

observed just preceding rapid tropopause ascent (>0.1 km h$^{-1}$). Symbol '*' labeled as





Tr1-Tr9 indicates the ending point of the corresponding trajectories in Fig.12b. (b)
Results of backward trajectories (colors in km) of the typical 9 selecting cases from
Fig.12a, providing the signature and source of possible stratospheric intrusions.





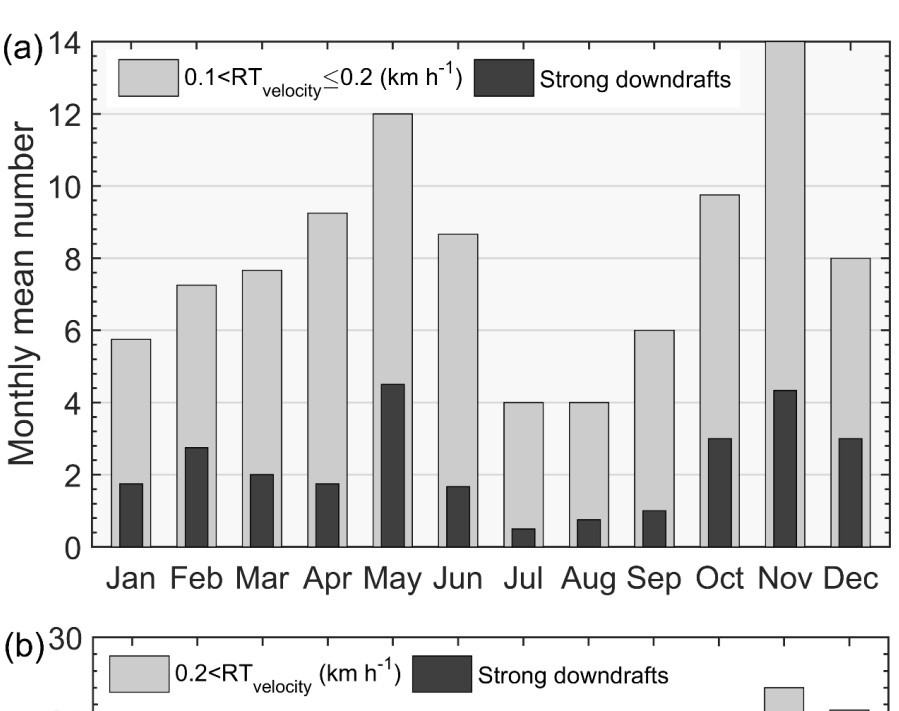

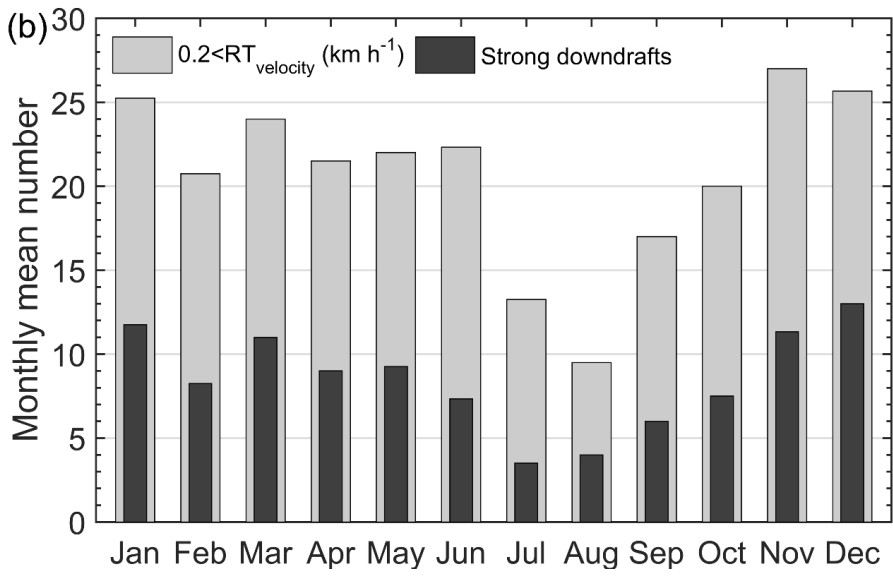


**Figure 13.** Four years (2012-2015) of radar-determined monthly mean number of rapid

tropopause ascent (gray bands) and the corresponding strong downdrafts just preceding

the rapid tropopause ascent (black bands). (a) Gray bands: with the ascent by at least

0.6 km and the excursion velocity is between 0.1-0.2 km h$^{-1}$; black bands: except for

the criteria of gray bands, strong downdrafts occurred preceding the rapid RT ascent



must exceed 0.5 m s$^{-1}$ and pass through the RT layer. (b) Same as (a) but for the
occasions when the ascent velocity is larger than 0.2 km h$^{-1}$. According to the study
here, the black bands in the histogram well represent the occurrence of possible
stratospheric intrusions.