# Peer review of "Strong downdrafts preceding rapid tropopause ascent and their potential to identify cross-tropopause stratospheric intrusions"

_Annales Geophysicae, 2018_

## Referee Comment (RC1) · Anonymous Referee #1 · 15 Aug 2018

Review of the manuscript "Strong downdrafts preceding rapid tropopause ascent and their potential to identify cross-tropopause stratospheric intrusions" Author(s): Feilong Chen et al. MS No.: angeo-2018-78

General Comments

The paper addresses an important aspect in the dynamics of the upper troposphere / lower stratosphere region, namely, relating tropopause height variability with stratospheric intrusions into the troposphere. Using VHF radar measurement data, including 3-D wind, the authors are able to diagnose events of considerable tropopause height drops. They additionally find that in a significant number of such cases, strong

downdrafts occur across the tropopause and extend to the mid and lower troposphere. These local downdrafts occur on the hourly time scale, and as such, they are revealed by the high-resolution radar data, but are missed by the coarser reanalysis or sporadic radiosondes.

My general concern is the interpretation of the considered events as 'rapid tropopause ascent', whereas the detailed case study (and many of the other cases presented) actually show a drop in tropopause height (such as occurring with a cutoff low, or an upper trough/PV streamer). The 'rapid ascent' seems to be a recovery from the drop in height, rather than the important phenomenon itself. Additionally, the downdrafts coincide with the lowest tropopause height and are related to the intrusions themselves. The reference to 'rapid tropopause ascent', i.e., higher tropopause height, compared to normal conditions, may give the opposite impression and confuse the readers. This notion appears in the title and throughout the text, and serves to identify the events climatologically using an ascent criterion, as shown in Fig. 13. In my opinion, diagnosing significant tropopause drops (i.e., both rapid descent and ascent) is more meaningful in the context of intrusions. It will be interesting to see how many of those are accompanied by strong downdrafts.

Overall, I found the presentation of the results in the text and the figures to be clear and concise. There are, however, some issues requiring further clarifications, and I therefore recommend publication if the general concern and the specific comments below are addressed.

Specific major comments

1. I do not understand how Figure 7 and the paragraph describing it in lines 283-287 help to relate the observed oscillations to the mountain waves. Please clarify, or delete this part (also from lines 407-409).

2. The trajectory analysis shows that the mid-tropospheric airmasses originate upstream from ∼7000-9000 m in altitude. This is commonly the upper troposphere, rather

than a clear stratospheric origin as stated (e.g., lines 312, 337). Please support the statements on the stratospheric origin by providing evidence of the lower tropopause height at those locations, or alternatively showing high PV values along the trajectories, or refrain from making these statements. It is relevant to note here that in Raveh-Rubin (2017), almost 99% of intrusions were not stratospheric in their origin.

Raveh-Rubin, S., 2017: Dry Intrusions: Lagrangian Climatology and Dynamical Impact on the Planetary Boundary Layer. J. Climate, 30, 6661–6682, https://doi.org/10.1175/JCLI-D-16-0782.1

Specific minor comments

1. Line 102-103: this sentence is unclear.

2. Lines 105-107: Please elaborate on the spatial and temporal relation between the tropopause ascent and the downward intrusions in Hocking et al. 2007.

3. L 148: "the characteristic (partial specular reflection) mentioned above" is unclear. Please clarify the characteristic (also unclear where is it mentioned above).

4. L 153-154: the description of the RT height determination should be written more clearly. Is it determined by searching upwards from 500 hPa for the first maximum of the gradient? It is unclear what "lower edge" or "secondary maximum" refer to.

5. L 247-248: please also refer to the very significant updrafts that follow the downdrafts. They can potentially be important for the recovery of the tropopause height back to normal, as they extend to the increasing height of the upper troposphere.

6. L 271: Is it related to the high winds at Q compared to P?

7. Lines 295-296: How does the figure support the cross-tropopause aspect?

8. Lines 297-304 and Fig. 9c. I suggest adding relative humidity to the profile, which may show clearer asymmetry between the east and western sides of the cutoff.

9. L 304-305: It is not clear if the low-level high PV is indeed stratospheric in origin as mentioned, or whether it is diabatically produced. See the distinction done in Škerlak et al 2015.

Škerlak, B., M. Sprenger, S. Pfahl, E. Tyrlis, and H. Wernli (2015), Tropopause folds in ERA‐Interim: Global climatology and relation to extreme weather events. J. Geophys. Res. Atmos., 120, 4860–4877. doi: 10.1002/2014JD022787.

10. L 354: Where are the high-pressure systems located relative to the events (height and horizontal location)?

11. Out of the 20 cases, it is a bit hard to keep track of their different characteristics. I suggest summarizing these in a table, and including the main features of Figures 12, S2 and the meteorological systems in lines 350-360.

12. L 363: I suggest to replace "predictor" by "diagnostic", as they occur at the same time. Also, delete "or prediction" from line 365.

13. Figure 1: It is suggested to add panels with sea-level pressure and low/mid tropospheric wind, to understand the environment of the downdrafts at these heights.

14. Figure 8 and in the main text: please add the time range of the satellite passage.

15. Figure 10 and 11 captions: please mention the height of the ending / starting point, respectively.

Technical Corrections

1. Velocities are shown in km/h and m/s throughout the manuscript. I suggest to be consistent and use only m/s.

2. Line 17: delete "possible", it is repeating after 'potential'.

3. L 22: delete "(weakened)", which is unclear in this context.

4. L 48: delete "long-term" from the second time it is mentioned, before 'seasonal'.

5. L 52: replace "when comes" to "with regards".

6. L 60: change "air transport" with "air is transported".

7. L 64: Move "although" to the beginning of the line.

8. L 93: delete "are".

9. L 97: change "comparing" to "compared to".

10. L 127: space is missing after the degree sign.

11. L 145: replace "to" with "away from".

12. L 159: replace "that" with ","

13. L 182: replace "with" with "interpolated into".

14. L 199: replace "bottom" to "southern tip".

15. L 200: add "as shown by the closed geopotential contour" after "site".

16. L 220: change "a" with "that".

17. L 221: change "didn't" with "did not".

18. L 223: replace "showed" with "shown".

19. L 224, and throughout the manuscript: change "UT" to "UTC".

20. L 263: delete "It is indeed reasonable.".

21. L 266: replace "impinges" with "impinging".

22. L 280: replace "Someone may be interested to notice" with "Interestingly".

23. L 298: delete "with".

24. L 320: Move "dominant" to after "flows".

25. L 328: "shown placed end-to-end" is unclear.
26. L 328: delete "and".

27. L 332: "four range gates" is unclear.

28. L 359: Add "(not shown)" after "48h".

29. L 360-361: delete "and not possible... satellite data", as it is redundant.

30. L 367: replace "have" with "has".

31. L 377: replace "excess" with "exceeds".

32. L 413: replace "What counts is" with "Yet".

33. L 414: add a '-' between "AIRS-retrieved"

34. L 415: add a '-' between "radar-derived"

35. Figure 6 legend: replace dotted orange line with a dashed line as in the plot itself.

36. Figure 12b: It is strongly suggested to use the same colour scale in all panels.

37. Figure S2: There are only 12 events presented, not 20. Please change the format of the dates in the panel titles to be the same as in Fig. 12.

---

## Referee Comment (RC2) · Anonymous Referee #2 · 21 Aug 2018

Comments of the revised manuscript entitled, 'Strong downdrafts preceding rapid tropopause ascent and their potential to identify cross-tropopause stratospheric intrusions' by Chen et al. The authors described 16 cases of stratospheric intrusion due to various synoptic cases over Beijing using MST radar. The results are supported by AIRS ozone observations along with ECMWF reanalysis and HYSPLIT back/forward trajectory model. This is certainly an interesting topic and the present scientific communities have an eye to understand the stratospheric intrusion and its impact on global ozone budget and earth's climate.

The manuscript is potential but need substantial revision before publication. Specific

points are following under :

(1) Title of the manuscript needs to change. How downdrafts can increase the tropopause level? It should be in other way, updrafts increase the tropopause level. Instead, as suitable tile can be chosen, something like : Rapid modulation of tropopause due to synoptic disturbances. . . . . . . . . . . (2) L19 : How authors define a strong updrafts, is it above 0.8 m/s. During many MST radar experiments, we observed vertical velocity up to $\pm12$ m/s. Anything above $\pm0.8$ m/s is considered to be presence of convective system but certainly not strong updrafts (which could be above $\pm2$ m/s). (3) L22 : 'destroyed' is not a correct word to use, instead 'stability of the tropopasue is weakened as observed by MST radar's SNR' (4) L25-27 : "According to . . . . . . .intrusions". This sentence is not necessary in the abstract. (5) L27-31 : "Twenty. . . . Discussed". These sentences can be combined and shorten. (6) L31-33: "The observations. . . . . . .observations". Authors cannot conclude. (7) L45 : How wind speeds plays a important role in STE? Is it shear generated turbulence ? (8) L48-50 : Sentence having repeating words. Few latest references are needed. (9) L64-66 : Sentence is not clear. Needs rephrasing. (10) L69-70: More recent references need to be included. For example, increase in surface ozone is observed during (a) mesoscale (Grant et al., 2008), and synoptic (Das et al., 2016; Jiang et al., 2015;) scale convective systems. (11) L84-86: As the sentence is written, 'Ozonesonde' is a tracer to detect the stratospheric intrusion. Sentence should be rewritten for better clarity. (12) L93 : It is too old to say that "Small scale intrusion are still remain uncertain" by referring Holten et al. (1995). There are many new research works and results are discovered in past 23 years. Authors must cite some latest references and what is the present scenario and lacuna in the existing recent literatures. (13) L94-96 : Unclear sentence. Needs to be rewritten. (14) L98 : Tropopause is not directly measured from VHF radar. There is an algorithm from which tropopause is detected using backscattering signal. Thus, author must caution, while describing about VHF radar capability. (15) L99-100 : "24 hours per day". This is not a scientific statement. Instead, author can write " VHF radar can be continuously used to detect tropopause height from

backscattering signal with an internal of 1 hour" (16) L101 : Reference is essential. (17) L102 : …...in many aspect……. : Authors must list few example. (18) L106 : 'radar-derived tropopause……...'. Along with the tropopause height, enhancement in the radar backscattering signal is essential to diagnosis the stratospheric intrusion. (19) L129 : '0.5 h time resolution' this is not the time resolution of radar measurements. It is the averaging time (post processing). (20) L141 : It is not correct to say that "strong potential temperature gradient". "Strong temperature inversion" is the correct word. (21) L156 : [2012] & [2014] (22) Under AIRS and ERA-I, proper citations are needed. (23) L180 : Replace 'Nov.2000' to 'November 2000', follow it throughout the manuscript. . (24) L212-215 : I do not agree with the statement. Generally, during deep convection, humidity increases. (25) L234 and Fig.5(d) : What does authors mean by 'Aspect sensitivity'. As I understood, it is the difference been zenith and off-zenith but what off-zenith angle and which direction? Is it 10-degree East ? (26) Fig.5 : (a), (b), (c), (d) should be level. (27) There is a huge difference between the radar detected and radiosonde detected tropopause. From 5, I could able to see about 2.5 km difference at 12 UTC on 30 November which is absolutely unacceptable (as the authors themselves have mentioned in the manuscript). I would like to suggest the author to relook on the algorithm for the detection of tropopause level by VHF radar (signal). Authors need to investigate further. (28) L241 : Needs reference. (29) L252-253 : I suggest to estimate CAPE (Convective Available Potential Energy) to confirm the occurrence of convection. (30) L259-260 : This statement is not fully correct. (31) L273-277 : larger value of aspect sensitivity cannot be from stratosphere, it is from stratified layers and attributed to the Fresnel reflection/scattering from sharp gradients in the radio refractive index. Thus, it will be mainly from the tropopause. I cannot understand how large value of aspect sensitivity indicates the stratospheric intrusion. If an isotropic turbulence persists, then aspect sensitivity will decrease. Needs further explanation. (32) L281 : replace 'Dec.' with 'December' and throughout the manuscript. (33) Fig.7 & L285-286 : I am confused, what actually authors wanted to discuss. Do they want to discuss convective or orographic (as authors mentioned presence of mountain of 1 km

north of radar site) generated gravity waves? If so, then it is not sufficient. Either authors need to make a separate section discussing on gravity wave structure (in-depth analysis) or omit this part. (34) L311-314 : From Fig.10, it is seen that the air masses is originated from 7-9 km at 40oN, which is upper troposphere. Thus, the statement of "air of stratospheric origin" is not correct or established here. Authors need to explain this analysis. (35) Fig.12 a: Quality of fig. is poor. Needs better clarity. (36) One interesting point I could able to find that whenever a synoptic event occurs, the tropopause height decreased to > 9 km (radar tropopause is much more lower to ∼ 6.5 km), which is a positive point to discuss in the manuscript. I think authors can put more stress in this point while discussing the back-trajectory analysis (see my previous comments). But again the question is that whether the tropopause height can be ∼6.5 km at 40oN? It is unacceptable fact, which again put a question on the algorithm used for detecting the tropopause height by MST radar. I again suggest authors to relook in this aspect (radar tropopause). (37) L407-410 : Mountain wave is no where discussed in the manuscript. See my previous comment. (38) Too many errors in English use, I do not list all that I found, but I hope the authors will carefully improve their writing.

References :

Jiang, Y. C., T. L. Zhao, J. Liu, X. D. Xu, C. H. Tan, X. H. Cheng, X. Y. Bi6, J. B. Gan, J. F. You, and S. Z. Zhao (2015), Why does surface ozone peak before a typhoon landing in southeast China? Atmos. Chem. Phys., 15, 13331–13338, doi:10.5194/acp-15-13331-2015

Grant, Deanne, Jose D. Fuentes, Marcia S. DeLonge, Stephen Chan, Everette Joseph, Paul Kucera, Seydi A. Ndiaye, Amadou T. Gaye (2008), Ozone transport by mesoscale convective storms in western Senegal, Atmos. Envir., 42, 7104–7114, doi:10.1016/j.atmosenv.2008.05.044

Das, S.S., M. V. Ratnam, K. N. Uma, K. V. Subrahmanyam, I.A.Girach, A. K. Patra,S. Aneesh, K.V. Suneeth, K. K. Kumar, A.P.Kesarkar, S. Sijikumar and G. Ramkumar

Influence of Tropical Cyclones on Tropospheric Ozone: Possible Implications (2016), Atmospheric Chemistry and Physics, 16, 4837-4847, doi : 10.5194/acp-16-1-2016

---

## Author Comment (AC1) · 25 Aug 2018

*Review of the manuscript "Strong downdrafts preceding rapid tropopause ascent and their potential to identify cross-tropopause stratospheric intrusions" Author(s): Feilong Chen et al.*

*MS No.: angeo-2018-78*

*General Comments*

**Response:** We really would like to thank the reviewer for giving us suggestions which help us to improve the quality of the paper. We have followed the reviewer's suggestions and the corresponding revision has been made.

*The paper addresses an important aspect in the dynamics of the upper troposphere / lower stratosphere region, namely, relating tropopause height variability with stratospheric intrusions into the troposphere. Using VHF radar measurement data, including 3-D wind, the authors are able to diagnose events of considerable tropopause height drops. They additionally find that in a significant number of such cases, strong downdrafts occur across the tropopause and extend to the mid and lower troposphere. These local downdrafts occur on the hourly time scale, and as such, they are revealed by the high-resolution radar data, but are missed by the coarser reanalysis or sporadic radiosondes.*

*My general concern is the interpretation of the considered events as 'rapid tropopause ascent', whereas the detailed case study (and many of the other cases presented) actually show a drop in tropopause height (such as occurring with a cutoff low, or an upper trough/PV streamer). The 'rapid ascent' seems to be a recovery from the drop in height, rather than the important phenomenon itself. Additionally, the downdrafts coincide with the lowest tropopause height and are related to the intrusions themselves. The reference to 'rapid tropopause ascent', i.e., higher tropopause height, compared to normal conditions, may give the opposite impression and confuse the readers. This notion appears in the title and throughout the text, and serves to identify the events climatologically using an ascent criterion, as shown in Fig. 13. In my opinion, diagnosing significant tropopause drops (i.e., both rapid descent and ascent) is more meaningful in the context of intrusions. It will be interesting to see how many of those are accompanied by strong downdrafts.*

**Response:** Your general concern is important and essential. In fact, tropopause drops, either slowly or rapidly, are close related to various synoptic-scale or mesoscale atmospheric processes such as cutoff low, low trough, or typhoon, which play an important role for potential stratospheric intrusions. However, not every such synoptic-scale or mesoscale atmospheric process is responsible for intrusions. On the other hand, the specific vertical velocity of the tropopause drop is most likely related to the strength of the corresponding atmospheric process, rather than the corresponding intrusion event. In other words, various atmospheric processes (and the accompanied tropopause drops) are important conditions for intrusions (or for the strong downdrafts in our study), but intrusion events are not close related to tropopause drops. As for the rapid ascent in tropopause height, no matter whether exists the tropopause drops, the potential intrusion events (intruded across the tropopause layer) will change the atmospheric structure. According to previous study by Hocking et al., 2007, the tropopause height started to ascent when the stratospheric air just intruded across the tropopause layer. In present study, the strong downdrafts and the accompanied rapid tropopause ascent (with specific erosion velocity) are found important features for the potential intrusions, although the ascent seems to be a recovery from the drop in tropopause height (many cases, not all). Therefore, we think the strong downdrafts just preceding the rapid tropopause ascent (black bands shown in Fig.13) may serve as a valuable predictor for possible stratospheric intrusions.

*Overall, I found the presentation of the results in the text and the figures to be clear and concise. There are, however, some issues requiring further clarifications, and I therefore recommend publication if the general concern and the specific comments below are addressed.*

*Specific major comments*
*1. I do not understand how Figure 7 and the paragraph describing it in lines 283-287 help to relate the observed oscillations to the mountain waves. Please clarify, or delete this part (also from lines 407-409).*
**Response:** Yes, you are right. We really thank you for the valuable comment and pointing out the deficiencies. Figure 7 and Figure S1 are indeed not essential and need to be deleted. The corresponding text and figures have been modified, please see the revised manuscript.
*2. The trajectory analysis shows that the mid-tropospheric airmasses originate upstream from _7000-9000 m in altitude. This is commonly the upper troposphere, rather than a clear stratospheric origin as stated (e.g., lines 312, 337). Please support the statements on the stratospheric origin by providing evidence of the lower tropopause height at those locations, or alternatively showing high PV values along the trajectories, or refrain from making these statements. It is relevant to note here that in Raveh-Rubin (2017), almost 99% of intrusions were not stratospheric in their origin.*
*Raveh-Rubin, S., 2017: Dry Intrusions: Lagrangian Climatology and Dynamical Impact on the Planetary Boundary Layer. J. Climate, 30, 6661–6682, https://doi.org/10.1175/JCLI-D-16-0782.1*
**Response:** Yes, you are right. Thank you very much for pointing out the deficiencies.

The statements of stratospheric intrusions are not appropriate for trajectory analysis. The further observations of the AIRS daily 500 hPa ozone distribution is essential to conclude that the intrusions are of stratospheric origin. The corresponding statements have been modified and replaced as "downward intrusions". Please see the corresponding text in the revised manuscript.

*Specific minor comments*

*1. Line 102-103: this sentence is unclear.*

**Response:** To make it more clear, we have added the expression "especially the criteria of identification by of radar observations" in the revised manuscript.

*2. Lines 105-107: Please elaborate on the spatial and temporal relation between the tropopause ascent and the downward intrusions in Hocking et al. 2007.*

**Response:** We have described the relation between the tropopause ascent and the downward intrusions in Hocking et al. 2007, please see the corresponding text "the RT height started to ascent when the stratospheric air just intruded across the tropopause layer."

*3. L 148: "the characteristic (partial specular reflection) mentioned above" is unclear. Please clarify the characteristic (also unclear where is it mentioned above).*

**Response:** To make the statement more clear, the corresponding sentence has been modified as "the characteristic (enhanced radar echoes due to partial specular reflection) mentioned above". This characteristic is mentioned above, the sentence "The tropopause, near which a strong potential temperature gradient exists, will lead to strong radar echoes in vertical incidence, as well as large radar aspect sensitivity (as shown in Figure 1)".

*4. L 153-154: the description of the RT height determination should be written more clearly. Is it determined by searching upwards from 500 hPa for the first maximum of the gradient? It is unclear what "lower edge" or "secondary maximum" refer to.*

**Response:** Yes, you are right. It is determined by searching upwards from 500 hPa for the first maximum of the gradient. We have rewritten the definition of RT height, please the sentence "Here, the radar-determined tropopause (RT) height is defined as the height (above 500 hPa) where the maximum vertical gradient of echo power located (shown as the orange circle in Figure 1a)." in the revised manuscript.

*5. L 247-248: please also refer to the very significant updrafts that follow the downdrafts. They can potentially be important for the recovery of the tropopause height back to normal, as they extend to the increasing height of the upper troposphere.*

**Response:** Thank you very much for the valuable comment. Our key point is the downdrafts followed by rapid RT ascent, no matter whether the ascent is the recovery normally or forced by the other factors such as the significant updrafts.

*6. L 271: Is it related to the high winds at Q compared to P?*

**Response:** The abnormal high radar aspect sensitivity is not related to the high winds at Q. High winds can't cause the difference of echo power between the vertical and oblique beams. As mentioned in the manuscript, the large value in radar aspect sensitivity is mainly caused by reflection from stable atmospheric layer, such as the tropopause or lower-stratosphere.

*7. Lines 295-296: How does the figure support the cross-tropopause aspect?*

**Response:** The cross-section of PV, humidity, and AIRS ozone clearly shown enhanced PV and ozone and dry air, which are typical characteristics of stratospheric air, intruded from lower stratosphere into the free troposphere.

*8. Lines 297-304 and Fig. 9c. I suggest adding relative humidity to the profile, which may show clearer asymmetry between the east and western sides of the cutoff.*

**Response:** The cross-section of relative humidity is shown below. Although it also shows obvious dry air intruded into the free troposphere, it is just similar to and not that better than the cross-section of specific humidity.

[Figure]

*9. L 304-305: It is not clear if the low-level high PV is indeed stratospheric in origin as mentioned, or whether it is diabatically produced. See the distinction done in Škerlak et al 2015.*

*Škerlak, B., M. Sprenger, S. Pfahl, E. Tyrlis, and H. Wernli (2015), Tropopause folds in ERAˇAˇ Interim: Global climatology and relation to extreme weather events. J. Geophys. Res. Atmos., 120, 4860–4877. doi: 10.1002/2014JD022787.*

**Response:** Indeed, we can not conclude that the low-level high PV is stratospheric in origin from cross-section of PV alone. Thus the cross-sections of humidity and ozone are presented to verify the stratospheric in origin.

*10. L 354: Where are the high-pressure systems located relative to the events (height and horizontal location)?*

**Response:** The low or high pressure systems are observed from 500 hPa meteorological chart. Please see the corresponding sentence "associated with low or high trough systems (at 500 hPa)".

*11. Out of the 20 cases, it is a bit hard to keep track of their different characteristics. I*

*suggest summarizing these in a table, and including the main features of Figures 12, S2 and the meteorological systems in lines 350-360.*

**Response:** We really would like to thank you for giving us the suggestion. The 20 cases identified in Fig. 12a are labeled as S1, S2, S3…, and S20, respectively. Their different characteristics, including background synoptic  condition, vertical velocity of  RT ascent, and 500 hPa ozone enhancement, have been summarized in a table (shown below). Please see the Table 2 in the revised manuscript.

Table 2. Characteristics of the 20 cases shown in Fig. 12a.

| Cases | Time (year/month/day) | Background  condition | Vertical velocity of  RT ascent | 500 hPa ozone enhancement |
|---|---|---|---|---|
| S1 | 2012/03/06 | Cut-off low | >0.2 km/h | Enhanced |
| S2 | 2012/03/06 | Cut-off low | >0.2 km/h | Enhanced |
| S3 | 2012/03/12 | Low/high trough | >0.2 km/h | Enhanced |
| S4 | 2012/03/13 | Low/high trough | >0.2 km/h | Enhanced |
| S5 | 2012/04/05 | Low/high trough | >0.2 km/h | Enhanced |
| S6 | 2012/04/05 | Low/high trough | >0.2 km/h | Enhanced |
| S7 | 2012/04/06 | Low/high trough | >0.2 km/h | Enhanced |
| S8 | 2012/06/13 | Cut-off low | >0.2 km/h | Enhanced |
| S9 | 2012/06/13 | Cut-off low | >0.2 km/h | Enhanced |
| S10 | 2013/08/02 | Cut-off low | >0.2 km/h | Enhanced |
| S11 | 2013/08/02 | Cut-off low | >0.2 km/h | Enhanced |
| S12 | 2013/08/03 | PV streamer | >0.2 km/h | Enhanced |
| S13 | 2013/08/03 | PV streamer | >0.2 km/h | Enhanced |
| S14 | 2014/01/02 | PV streamer | >0.2 km/h | None |
| S15 | 2014/01/02 | PV streamer | >0.2 km/h | None |
| S16 | 2014/01/03 | PV streamer | 0.1-0.2 km/h | None |
| S17 | 2014/01/04 | Low/high trough | >0.2 km/h | None |
| S18 | 2014/05/02 | Low/high trough | 0.1-0.2 km/h | Enhanced |
| S19 | 2014/05/02 | Low/high trough | >0.2 km/h | Enhanced |
| S20 | 2015/01/03 | PV streamer | >0.2 km/h | None |

*12. L 363: I suggest to replace "predictor" by "diagnostic", as they occur at the same time. Also, delete "or prediction" from line 365.*

**Response:** Thank you very much for pointing out the deficiencies. The "predictor" has been replaced by "diagnostic", and the "or prediction" been deleted.

*13. Figure 1: It is suggested to add panels with sea-level pressure and low/mid tropospheric wind, to understand the environment of the downdrafts at these heights.*

**Response:** We really would like to thank you for giving us the suggestion. We consider that the 500 hPa geopotential height, Time series of surface hourly meteorological measurements, and maps of Outgoing Longwave Radiation are

enough to understand the environment of the downdrafts.

*14. Figure 8 and in the main text: please add the time range of the satellite passage.*

**Response:** The AIRS retrieved data of daytime ascending pass (south pole to north pole) are used in Fig. 8 (Fig. 7 in the revised manuscript). According to the Aqua Orbit Tracks, as shown below, the time range of the satellite passage is between ~04:00-07:25 on 30 November and between ~03:15-06:35. The corresponding text has been added in the revised manuscript.

[Figure]

30 November 2014

1 December 2014

*15. Figure 10 and 11 captions: please mention the height of the ending / starting point, respectively.*

**Response:** Thank you very much for pointing out the deficiencies. the ending / starting point has been mentioned in the corresponding figure captions. Please see figure 9 and 10 in the revised manuscript.

*Technical Corrections*

*1. Velocities are shown in km/h and m/s throughout the manuscript. I suggest to be consistent and use only m/s.*

**Response:** Thank you for pointing out the suggestion. The km/h is used for radar tropopause ascent, and m/s is mainly used for 3D winds. We considered that they are reasonable.

*2. Line 17: delete "possible", it is repeating after 'potential'.*

**Response:** We really thank the **referee** for pointing out the deficiencies. We have modified the corresponding text.

*3. L 22: delete "(weakened)", which is unclear in this context.*

**Response:** We have modified the corresponding text.

*4. L 48: delete "long-term" from the second time it is mentioned, before 'seasonal'.*

**Response:** We have modified the corresponding text.

*5. L 52: replace "when comes" to "with regards".*

**Response:** We have replaced "when comes" to "with regards".

*6. L 60: change "air transport" with "air is transported".*

**Response:** The corresponding text has been changed.

*7. L 64: Move "although" to the beginning of the line.*

**Response:** We have modified the corresponding text.

*8. L 93: delete "are".*

**Response:** The corresponding text has been deleted.

*9. L 97: change "comparing" to "compared to".*

**Response:** The corresponding text has been changed.

*10. L 127: space is missing after the degree sign.*

**Response:** The space has been added in the corresponding text.

*11. L 145: replace "to" with "away from".*

**Response:** We have replaced "to" with "away from".

*12. L 159: replace "that" with ",".*

**Response:** We have replaced "that" with ",".

*13. L 182: replace "with" with "interpolated into".*

**Response:** We have replaced "with" with "interpolated into".

*14. L 199: replace "bottom" to "southern tip".*

**Response:** We have replaced "bottom" to "southern tip".

*15. L 200: add "as shown by the closed geopotential contour" after "site".*

**Response:** The corresponding text has been added.

*16. L 220: change "a" with "that".*

**Response:** The corresponding text has been changed.

*17. L 221: change "didn't" with "did not".*

**Response:** The corresponding text has been changed.

*18. L 223: replace "showed" with "shown".*

**Response:** We have replaced "showed" with "shown".

*19. L 224, and throughout the manuscript: change "UT" to "UTC".*

**Response:** We have replaced "UT" with "UTC" throughout the manuscript.

*20. L 263: delete "It is indeed reasonable.".*

**Response:** The corresponding text has been deleted.

*21. L 266: replace "impinges" with "impinging".*

**Response:** The corresponding text has been changed.

*22. L 280: replace "Someone may be interested to notice" with "Interestingly".*

**Response:** The corresponding text has been changed.

*23. L 298: delete "with".*

**Response:** The corresponding text has been deleted.

*24. L 320: Move "dominant" to after "flows".*

**Response:** The corresponding text has been changed.

*25. L 328: "shown placed end-to-end" is unclear.*

**Response:** The time is not continuous and is placed end-to-end with intervals of 2.5 hours (white field).

*26. L 328: delete "and".*

**Response:** The corresponding text has been deleted.

*27. L 332: "four range gates" is unclear.*

**Response:** One range gate indicates the height resolution of the MST radar (150 m).

*28. L 359: Add "(not shown)" after "48h".*

**Response:** The corresponding text has been changed.

*29. L 360-361: delete "and not possible: : : satellite data", as it is redundant.*

**Response:** The corresponding text has been deleted.

*30. L 367: replace "have" with "has".*

**Response:** The corresponding text has been changed.

*31. L 377: replace "excess" with "exceeds".*

**Response:** The corresponding text has been changed.

*32. L 413: replace "What counts is" with "Yet".*

**Response:** The corresponding text has been changed.

*33. L 414: add a '-' between "AIRS-retrieved"*

**Response:** The corresponding text has been changed.

*34. L 415: add a '-' between "radar-derived"*

**Response:** The corresponding text has been changed.

*35. Figure 6 legend: replace dotted orange line with a dashed line as in the plot itself.*

**Response:** The dotted orange line has replaced with dashed line (legend).

[Figure]

*36. Figure 12b: It is strongly suggested to use the same colour scale in all panels.*

**Response:** Thank you very much for pointing out the suggestion. All the panels in Fig. 12b have modified and using the same colour scale. Please see Fig. 11b in the revised manuscript.

[Figure]

*37. Figure S2: There are only 12 events presented, not 20. Please change the format of the dates in the panel titles to be the same as in Fig. 12.*
**Response:** The format of the dates in the panel titles of Fig. S2 has been modified. Please see Fig. S1 in the revised manuscript.

---

## Author Comment (AC3) · 25 Aug 2018

**Supplementary material**

It is well established that AIRS retrieved ozone provides strong information for identifying a stratospheric intrusion event. Supplementary figure S2 shows the AIRS derived daily 500 hPa ozone distribution corresponding to the cases shown in Figure 12a, to assist to support the evidence of possible stratospheric intrusions closely associated with the strong downdrafts preceding RT ascent. Due to some of the cases occur close together, as well as due to the limited resolution in AIRS retrieved data (daily in fact), not every individual case in Figure 12a can be clearly checked from AIRS observations. Almost every case (except for the cases labeled by A, B, C, and D) is associated with some form of significant 500 hPa ozone enhancement, indicating the intrusions of stratospheric origin.

[Figure]

**Supplementary figure S1.** AIRS retrieved daily 500 hPa ozone distribution along with

the tropopause height contour corresponding to the cases identified in Figure 11a.

The black star shows the location of the Beijing MST radar.

---

## Author Comment (AC4) · 29 Aug 2018

*Comments of the revised manuscript entitled, 'Strong downdrafts preceding rapid tropopause ascent and their potential to identify cross-tropopause stratospheric intrusions' by Chen et al.*

*The authors described 16 cases of stratospheric intrusion due to various synoptic cases over Beijing using MST radar. The results are supported by AIRS ozone observations along with ECMWF reanalysis and HYSPLIT back/forward trajectory model. This is certainly an interesting topic and the present scientific communities have an eye to understand the stratospheric intrusion and its impact on global ozone budget and earth's climate. The manuscript is potential but need substantial revision before publication.*

**Response:** We really would like to thank the reviewer for giving us constructive and supportive suggestions which would help us to improve the quality of the paper. We also really would like to thank the reviewer for pointing out our deficiencies. We have followed the reviewer's suggestion and substantial revisions have been made in the revised manuscript. We have repeated the comments of the reviewer in italics before our response. The revised manuscript with tracked changes (highlighted in red font) is provided.

*Specific points are following under:*

*(1) Title of the manuscript needs to change. How downdrafts can increase the tropopause level? It should be in other way, updrafts increase the tropopause level. Instead, as suitable tile can be chosen, something like : Rapid modulation of tropopause due to synoptic disturbances: : :: : :: : :: : :.*

**Response:** Thank you very much for pointing out the comment. But, our attention have been focused on the key point that the strong downdrafts preceding the rapid tropopause (downdrafts occurred just before the rapid radar tropopause ascent) ascent have a strong potential for identifying possible intrusions. We didn't show and indicate that the downdrafts can increase the tropopause level. According to a detailed case and other 20 cases, observations have shown that the downdrafts followed by rapid radar tropopause ascent can be used to infer the occurrence of intrusions of stratospheric origin.

*(2) L19 : How authors define a strong updrafts, is it above 0.8 m/s. During many MST radar experiments, we observed vertical velocity up to _12 m/s. Anything above _0.8 m/s is considered to be presence of convective system but certainly not strong updrafts (which could be above _2 m/s).*

**Response:** We think the reviewer refers to the definition of strong downdrafts, which occurred just before the rapid RT ascent and is the key point. Three criteria have been proposed in the manuscript to define the case of strong downdrafts followed by rapid RT ascent. That is: 1) the amplitude of the RT ascent should exceed 0.6 km (four range

gates), 2) vertical velocities of the RT ascent excess 0.1 km/h, 3) the downdrafts occurred preceding the RT ascent should >0.5 m/s, and the height region of the downdrafts should pass through the RT layer. The background weather condition have been discussed from 500 hPa geopotential height, time series of surface hourly meteorological measurements, and maps of Outgoing Longwave Radiation. Besides, no matter where the downdrafts accurately originate, we indeed found that the strong downdrafts preceding rapid RT ascent can be used as a valuable predictor for possible stratospheric intrusions.

*(3) L22 : 'destroyed' is not a correct word to use, instead 'stability of the tropopasue is weakened as observed by MST radar's SNR'*

**Response:** Yes, you are right. Thank you very much for pointing out the deficiencies. We have followed the reviewer's suggestion and changed the associated sentence in the revised manuscript. It has changed to "Within the height region of the downdrafts, the stability of the radar tropopause seems to be weakened".

*(4) L25-27 : "According to : : :: : :.intrusions". This sentence is not necessary in the abstract.*

**Response:** We really would like to thank the reviewer for pointing out the deficiency. The corresponding sentence has been deleted.

*(5) L27-31 : "Twenty: : :. Discussed". These sentences can be combined and shorten.*

**Response:** Thank you for pointing out the comment. We consider that the two sentences need to be separated. The former one introduce the other 20 cases and their relationship with the occurrence of possible stratospheric intrusions. The latter one is about the statistics of the downdrafts.

*(6) L31-33: "The observations: : :: : :.observations". Authors cannot conclude.*

**Response:** We really thank the reviewer for the valuable comment. A major result of our study is the observations of strong downdrafts just preceding the rapid tropopause ascent, which serve as a valuable predictor for possible stratospheric intrusions. This potential value is verified from a detailed case and other 20 typical cases.

*(7) L45 : How wind speeds plays a important role in STE? Is it shear generated turbulence ?*

**Response:** Thank you for pointing out the comment. The sentence "Consequently, the natural stable tropopause layer, characterized by strong gradients of trace constituents and wind speeds, plays an important role in stratosphere-troposphere exchange (STE) processes" means that the stable tropopause layer plays an important role in STE

processes, rather than the wind speeds. The tropopause layer is a significant barrier for STE (Mahlman, 1997).

Mahlman, J. D.: Dynamics of transport processes in the upper troposphere. Science, 276(5315), 1079-1083, 1997.

*(8) L48-50 : Sentence having repeating words. Few latest references are needed.*

**Response:** Yes, you are right. Thank you very much for pointing out the deficiencies. The repeating word is "long-term", we have changed the sentence to "From a long-term point of view, the seasonal variation of the tropopause height determines the seasonal variation of the flux of stratospheric air into the free troposphere (Appenzeller et al., 1996)".

*(9) L64-66 : Sentence is not clear. Needs rephrasing.*

**Response:** Thank you for pointing out the comment. The corresponding sentence has been changed to "Although photochemical production within the troposphere is the main source of tropospheric ozone, the influence of the downward stratospheric intrusions on the tropospheric ozone content cannot be ignored (Oltmans and Levy II, 1992; Monks, 2000; Stevenson et al., 2006)".

*(10) L69-70: More recent references need to be included. For example, increase in surface ozone is observed during (a) mesoscale (Grant et al., 2008), and synoptic (Das et al., 2016; Jiang et al., 2015;) scale convective systems.*

**Response:** We have followed the reviewer's suggestion, and three more references have been added. Please see the corresponding sentence "sometimes even deep to the surface (e.g. Gerasopoulos et al., 2006; Grant et al., 2008; Lefohn et al., 2011; Jiang et al., 2015; Das et al., 2016;)" in the revised manuscript.

*(11) L84-86: As the sentence is written, 'Ozonesonde' is a tracer to detect the stratospheric intrusion. Sentence should be rewritten for better clarity.*

**Response:** Yes, you are right. The corresponding sentence has been changed from "Among them, balloon-borne ozonesonde sounding are without doubt one of the most appropriate tools, but is limited by coverage (He et al., 2011) and not possible to obtain continuous profiles with fine temporal resolution" to "Balloon-borne ozonesonde sounding is an effective tool to make measurements of ozone with high vertical resolution, but is limited by coverage (He et al., 2011) and temporal resolution."

*(12) L93 : It is too old to say that "Small scale intrusion are still remain uncertain" by referring*

*Holten et al. (1995). There are many new research works and results are discovered in past 23 years. Authors must cite some latest references and what is the present scenario and lacuna in the existing recent literatures.*

**Response:** Yes, you are right. To make it more appropriate, the corresponding sentence ha been changed to "By far, large-scale STE has been widely studied and is fairly well understood, but the details of small scale intrusions still need more researches (e.g. Holton et al., 1995)" in the revised manuscript.

*(13) L94-96 : Unclear sentence. Needs to be rewritten.*

**Response:** We have followed the reviewer's suggestion and changed the corresponding sentence to "Kumar and Uma (2009) reported that the shortage of direct measurements of vertical winds near the tropopause may be responsible for the lack of fine-scale observations of smaller scale intrusions".

*(14) L98 : Tropopause is not directly measured from VHF radar. There is an algorithm from which tropopause is detected using backscattering signal. Thus, author must caution, while describing about VHF radar capability.*

**Response:** Yes, you are right. The tropopause height is not measured directly from MST radar, it is retrieved from radar backscattered echo power in vertical incidence. The corresponding sentence has been changed to "Very-High-Frequency (VHF) radars, compared to the tools mentioned above, are capable of continuously monitoring the atmosphere under any weather conditions and detecting tropopause height from backscattered signal with both high temporal and spatial resolution". The details of the definition of radar tropopause is presented in section 2.1.

*(15) L99-100 : "24 hours per day". This is not a scientific statement. Instead, author can write " VHF radar can be continuously used to detect tropopause height from backscattering signal with an internal of 1 hour"*

**Response:** Thank you very much for pointing out the deficiency. The corresponding sentence has been changed to "Very-High-Frequency (VHF) radars, compared to the tools mentioned above, are capable of continuously monitoring the atmosphere under any weather conditions and detecting tropopause height from backscattered signal with both high temporal and spatial resolution".

*(16) L101 : Reference is essential.*

**Response:** Thank you very much for pointing out the deficiency. Two references have been added in the revised manuscript. Please see the sentence "During the past two

decades, VHF radar measurements were commonly used to assist to study the stratospheric intrusions (e.g. Hocking et al., 2007; Das et al., 2016)."

*(17) L102 : : : :..in many aspect: : :: : : : Authors must list few example.*

**Response:** To make it more clear, we have changed the corresponding sentence to "However, it still remains uncertain in many aspects when using only the VHF radar to identify intrusion events, especially the criteria for the identification".

*(18) L106 : 'radar-derived tropopause: : :: : :..'. Along with the tropopause height, enhancement in the radar backscattering signal is essential to diagnosis the stratospheric intrusion.*

**Response:** The corresponding sentence has been changed to "They found that the rapid ascent in RT altitude (>0.2 km/h) can be a valuable diagnostic for possible stratospheric intrusions and the RT height started to ascent just when the stratospheric air intruded across the tropopause layer".

*(19) L129 : '0.5 h time resolution' this is not the time resolution of radar measurements. It is the averaging time (post processing).*

**Response:** In order to achieve simultaneously measurements from troposphere, lower stratosphere, and mesosphere, the radar is designed to operate in three separate modes: low mode (designed range 2.5-~12 km), middle mode (10-~25 km), and high mode (60-~90 km) with height resolutions of 150, 600, and 1200 m, respectively. Under the daily routine normal operation, 15-min break is followed by the 15-min operation cycle (5 min for each mode). Indeed, this 5 min operation for each mode in 5 beams (providing a profile of radar measurements) is the averaging time after post processing. However, the time resolutions of the low, middle, and high mode measurements are all 0.5 h.

*(20) L141 : It is not correct to say that "strong potential temperature gradient". "Strong temperature inversion" is the correct word.*

**Response:** Thank you for pointing out the comment. In fact, we meant the strong inversion in potential temperature.

*(21) L156 : [2012] & [2014]*

**Response:** Thank you very much for pointing out the deficiencies. The corresponding text have been corrected.

*(22) Under AIRS and ERA-I, proper citations are needed.*

**Response:** Proper citation has been added in the corresponding text.

*(23) L180 : Replace 'Nov.2000' to 'November 2000', follow it throughout the manuscript. .*

**Response:** Thank you very much for pointing out the deficiencies. We have followed the reviewer's comment and changed the corresponding text throughout the manuscript.

*(24) L212-215 : I do not agree with the statement. Generally, during deep convection, humidity increases.*

**Response:** Thank you for pointing out the comment. We actually meant the rapid decrease in humidity under the background condition of cut-off low system, rather than the potential deep convection.

*(25) L234 and Fig.5(d) : What does authors mean by 'Aspect sensitivity'. As I understood, it is the difference been zenith and off-zenith but what off-zenith angle and which direction? Is it 10-degree East ?*

**Response:** Yes, you are right. The Aspect sensitivity here means the difference in backscattered echo power between zenith and off-zenith beam. In the manuscript, the radar aspect sensitivity is expressed as the ratio between vertical and oblique (here used the 15-degree north) beam echo power. Please see the sentence "In addition, the radar aspect sensitivity, expressed as the ratio between vertical ($p_v$) and oblique ($p_o$, here used the 15-degree north) beam echo power, is mainly caused by the horizontally stratified anisotropic stable air and thus will be used as potential signature of stratospheric intrusions in the troposphere (e.g. Kim et al., 2001)" in the revised manuscript.

*(26) Fig.5 : (a), (b), (c), (d) should be level.*

**Response:** We have followed the reviewer's suggestion and modified the (a), (b), (c), (d) to be level. Please see the picture below.

[Figure]

*(27) There is a huge difference between the radar detected and radiosonde detected tropopause. From 5, I could able to see about 2.5 km difference at 12 UTC on 30 November which is absolutely unacceptable (as the authors themselves have mentioned in the manuscript). I would like to suggest the author to relook on the algorithm for the detection of tropopause level by*

*VHF radar (signal). Authors need to investigate further.*

**Response:** Large difference in height between the radar detected and radiosonde detected tropopause is commonly observed, especially during severe weather conditions. To make the definition of radar tropopause more clear, the sentence about the definition has changed to "Here, the radar-determined tropopause (RT) height is defined as the height (above 500 hPa) where the maximum vertical gradient of echo power located (shown as the orange circle in Figure 1a)" in the revised manuscript. Also shown below is the altitude-time intensity plot of (a) radar backscattered echo power and (b) radar aspect sensitivity for February 2014. The tropopauses determined based on the radar echo definition are shown as a black solid curve. The red crosses and purple dots indicate the location of the Lapse-rate (LRT) and 2PVU tropopause (PVT), respectively. In general, the RT agreed well with both the LRT. However, it is clear to be seen that the differences between the RT and LRT are large (reach to ~1-2 km) on some days especially when the RT experience rapid change (severe weather conditions). The difference of the definitions themselves is to a large degree the main contributing factor. Such as for the cases on 4 and 5 February 2012 when large changes in RT height occurred, a second layer with significantly enhanced echo power is observed above the RT and its altitude of maximum echo power gradient is just well consistent with the LRT (a). According to the definition, the RT well matched the lower part but the LRT often matched the upper part, similar to that observed by Yamamoto et al., (2003) and Fukao et al., (2003).

[Figure]

*Yamamoto, M.K., Oyamatsu, M., Horinouchi, T., Hashiguchi, H., Fukao, S., (2003). High time resolution determination of the tropical tropopause by the Equatorial Atmosphere Radar. Geophys. Res. Lett. 30 (21), 2094. http://dx.doi.org/10.1029/ 2003GL018072.*

*Fukao, S., H. Hashiguchi, M. Yamamoto, T. Tsuda, T. Nakamura, M. K. Yamamoto, T. Sato, M. Hagio, and Y. Yabugaki, Equatorial Atmosphere Radar (EAR): System description and first results, Radio Sci., 38(3), 1053, doi:10.1029/2002RS002767, 2003.*

*(28) L241 : Needs reference.*

**Response:** We have followed the reviewer's suggestion and added two references in the corresponding sentence. Please see sentence "It is the difference in definition that contribute most to the large differences, especially under the tropopause fold conditions (e.g. Yamamoto et al., 2003 and Fukao et al., 2003)." in the revised manuscript.

*Yamamoto, M., Oyamatsu, M., Horinouchi, T., Hashiguchi, H., & Fukao, S.: High time resolution determination of the tropical tropopause by the Equatorial Atmosphere Radar. Geophysical Research Letters, 30(21), 2003.*

*Fukao, S., H. Hashiguchi, M. Yamamoto, T. Tsuda, T. Nakamura, M. K. Yamamoto, T. Sato, M. Hagio, and Y. Yabugaki.: Equatorial Atmosphere Radar (EAR): System description and first results, Radio Sci., 38(3), 1053, doi:10.1029/2002RS002767, 2003.*

*(29) L252-253 : I suggest to estimate CAPE (Convective Available Potential Energy) to confirm the occurrence of convection.*

**Response:** We really would like to thank you for giving us the suggestion. We consider

that the 500 hPa geopotential height, time series of surface hourly meteorological measurements, and maps of Outgoing Longwave Radiation are enough to understand the background condition.

*(30) L259-260 : This statement is not fully correct.*

**Response:** Thank you very much for pointing out the deficiency. The corresponding sentence has been changed to "The research by Hocking et al. (2007) has suggested that the rapid ascent in RT height (>0.2 km h-1) can be a valuable predictor for the occurrence of stratospheric intrusions" in the revised manuscript.

*(31) L273-277 : larger value of aspect sensitivity cannot be from stratosphere, it is from stratified layers and attributed to the Fresnel reflection/scattering from sharp gradients in the radio refractive index. Thus, it will be mainly from the tropopause. I cannot understand how large value of aspect sensitivity indicates the stratospheric intrusion. If an isotropic turbulence persists, then aspect sensitivity will decrease. Needs further explanation.*

**Response:** Yes, you are right. We mentioned that the abnormal large aspect sensitivity may be a weak signature of the possible intrusions. We have explained that in normal conditions, the aspect sensitivity is usually low in value in the troposphere, due to the presence of isotropic turbulence. The large value in radar aspect sensitivity is mainly caused by reflection from stable atmospheric layer, such as the tropopause or lower-stratosphere. When stable stratospheric air intrudes into the troposphere and without mixing with the surrounding air mass, the intrusions in the free troposphere will be reflected as abnormal large aspect sensitivity. Thus, it is just a potential evidence of possible intrusions. Further strong evidence of the relevant intrusions in dynamical and chemical aspects are presented using satellite AIRS and global reanalysis data.

*(32) L281 : replace 'Dec.' with 'December' and throughout the manuscript.*

**Response:** Thank you very much for pointing out the deficiencies. We have followed the reviewer's comment and changed the corresponding text throughout the manuscript.

*(33) Fig.7 & L285-286 : I am confused, what actually authors wanted to discuss. Do they want to discuss convective or orographic (as authors mentioned presence of mountain of 1 km north of radar site) generated gravity waves? If so, then it is not sufficient. Either authors need to make a separate section discussing on gravity wave structure (in-depth analysis) or omit this part.*

**Response:** Yes, you are right. We really thank you for the valuable comment and pointing out the deficiencies. Figure 7 and Figure S1 are indeed not essential and need

to be deleted. The corresponding text and figures have been modified and deleted, please see the revised manuscript.

*(34) L311-314 : From Fig.10, it is seen that the air masses is originated from 7-9 km at 40oN, which is upper troposphere. Thus, the statement of "air of stratospheric origin" is not correct or established here. Authors need to explain this analysis.*

**Response:** Yes, you are right. Thank you very much for pointing out the deficiencies. The statements of stratospheric intrusions are not appropriate for trajectory analysis. The further observations of the AIRS daily 500 hPa ozone distribution is essential to further verify the intrusions are of stratospheric origin. The corresponding statements have been modified and replaced as "downward intrusions". Please see the corresponding text in the revised manuscript.

*(35) Fig.12 a: Quality of fig. is poor. Needs better clarity.*

**Response:** Thank you very much for pointing out the suggestion. All the panels in Fig. 12 have modified. Please see Fig. 11b in the revised manuscript. To make the cases identified in Figure 11a more clear, the cases are labeled as S1, S2, S3, …, and S20. The figure is shown below.

[Figure]

*(36) One interesting point I could able to find that whenever a synoptic event occurs, the tropopause height decreased to > 9 km (radar tropopause is much more lower to _ 6.5 km), which is a positive point to discuss in the manuscript. I think authors can put more stress in this point while discussing the back-trajectory analysis (see my previous comments). But again the question is that whether the tropopause height can be _6.5 km at 40oN? It is unacceptable fact, which again put a question on the algorithm used for detecting the tropopause height by MST radar. I again suggest authors to relook in this aspect (radar tropopause).*

**Response:** According to our previous response. Large difference in height between the radar detected and radiosonde detected tropopause is commonly observed, especially during severe weather conditions. To make the definition of radar tropopause more clear, the sentence about the definition has changed to "Here, the radar-determined tropopause (RT) height is defined as the height (above 500 hPa) where the maximum

vertical gradient of echo power located (shown as the orange circle in Figure 1a)" in the revised manuscript. Shown below is the altitude-time intensity plot of radar backscattered echo power and radar aspect sensitivity for February 2014. Differences between the RT and LRT are large (reach to ~1-2 km) on some days especially when the RT experience rapid change (severe weather conditions). The difference of the definitions themselves is to a large degree the main contributing factor. Such as for the cases on 4 and 5 February 2012 when large changes in RT height occurred, a second layer with significantly enhanced echo power is observed above the RT and its altitude of maximum echo power gradient is just well consistent with the LRT. According to the definition, the RT well matched the lower part but the LRT often matched the upper part, similar to that observed by Yamamoto et al., (2003) and Fukao et al., (2003). In the manuscript, such difference is also discussed from Figure 6. At 40°N, conditions with tropopause height (both the LRT and the RT) lower than 6.5 km is rare, but it actually exist, no matter whether the difference between the LRT and the RT is large.

*(37) L407-410 : Mountain wave is no where discussed in the manuscript. See my previous comment.*

**Response:** Yes, you are right. Thank you very much for pointing out the deficiencies. The discussion about the mountain wave is actually redundant. The corresponding text and figures have been modified and deleted, please see the revised manuscript.

(38) Too many errors in English use, I do not list all that I found, but I hope the authors will carefully improve their writing.

**Response:** We really would like to thank you for pointing out our deficiencies. We also very sorry for our poor English writing that makes you difficult to read. In the revised manuscript, large revisions have been made, including the issues about the English writing.

References :

Jiang, Y. C., T. L. Zhao, J. Liu, X. D. Xu, C. H. Tan, X. H. Cheng, X. Y. Bi6, J. B. Gan, J. F. You, and S. Z. Zhao (2015), Why does surface ozone peak before a typhoon landing in southeast China? Atmos. Chem. Phys., 15, 13331–13338, doi:10.5194/acp-15- 13331-2015

Grant, Deanne, Jose D. Fuentes, Marcia S. DeLonge, Stephen Chan, Everette Joseph, Paul Kucera, Seydi A. Ndiaye, Amadou T. Gaye (2008), Ozone transport by mesoscale convective storms in western Senegal, Atmos. Envir., 42, 7104–7114,

doi:10.1016/j.atmosenv.2008.05.044

Das, S.S., M. V. Ratnam, K. N. Uma, K. V. Subrahmanyam, I.A.Girach, A. K. Patra,S. Aneesh, K.V. Suneeth, K. K. Kumar, A.P.Kesarkar, S. Sijikumar and G. Ramkumar Influence of Tropical Cyclones on Tropospheric Ozone: Possible Implications (2016), Atmospheric Chemistry and Physics, 16, 4837-4847, doi : 10.5194/acp-16-1-2016

---

## Author Response (AR1)

*Topical Editor Decision: Publish subject to revisions (further review by editor and referees) (29 Aug 2018) by Marc Salzmann*

*Comments to the Author:*

*Thank you very much for including a revised manuscript in your response to the reviewers. It would have been enough to respond to the comments first and later submit a revised manuscript. But of course, a revised manuscript can help to make your points.*

**Dear Editor:** Thank you very much for your kindly comments and valuable suggestions, which help us to improve the quality of the paper. We have followed the editor's suggestions and the corresponding revision has been made (on the basis of the revised version according to reviewer #1 and reviewer #2). The changes we made are shown in red font. The revised manuscript with tracked changes is attached later.

*Based on your responses and the revised manuscript, my impression is that major comment #1 by reviewer number #1 and point #1 by reviewer #2 should be addressed by additional changes in the in the manuscript. The motivation for looking at ascents needs to be better explained in the manuscript. For example, you could include at least one or two sentences based on your response to reviewer #1 around line 104. If the motivation for your approach is made clear in the manuscript, personally I would not insist on changing the title, since it seems to express what you are intending to say. In general, however, it is often good to clarify things that are unclear in the manuscript, since not all readers may refer to the public discussion. Please note that there will be another opportunity to submit a revised manuscript and that this revised manuscript will be sent out for another round of reviews.*

**Response**: To make our points more clear, additional changes related to the choice of ascents are actually essential in the manuscript. According to your suggestions, changes have been made in the corresponding text. Please see lines 104-112, that is sentences "The research by Hocking et al., (2007) have achieved a development in this issue and reported that the rapid ascent in RT altitude (>0.2 km/h) can be a valuable diagnostic for possible stratospheric intrusions. They observed the RT height started to ascent just when the stratospheric air intruded across the tropopause layer directly, although the ascent seems to be a recovery from the drop in tropopause height (many cases, not all, including this study). On the other hand, in fact, tropopause drops are more close related to various atmospheric processes such as cutoff low and low/high trough, rather than the corresponding intrusion process itself. Therefore, tropopause ascent is one of the key objects in this study."

*I also noted that the first sentence of your acknowledgment seems to imply that your case study qualifies for the criterion in Raveh-Rubin (2017). If it is so, this should be explained (including a few details) in the main body of the manuscript and not in the acknowledgment section and there should definitely be a proper citation of Raveh-Rubin (2017).*

**Response:** Yes, Prof Raveh-Rubin helped us to check the case study using Lagrangian method, the case indeed qualifies for the criterion in Raveh-Rubin (2017). She found a large dry intrusion associated with the case of cut-off low. This has been mentioned in the revised manuscript, please see Lines 310-312, and a proper citation of Raveh-Rubin (2017) has been added. However, we must admit that we do not familiar with the Lagrangian method and criterion in Raveh-Rubin (2017).

*Non-public comments to the author:*

*I will be on vacation until 18 September, so please take your time and excuse me should it take me longer than usual to take a decision in case you submit the manuscript very soon.*

**Response:** Thank you very much again for your kindly and valuable comments. Also thank you very much for your quick decision. Personally, this paper is very important for me, because it relates to my doctor graduate directly.

[revised manuscript text omitted]

---

## Author Response (AR2)

**Topical Editor Decision: Publish subject to minor revisions (review by editor)** (05 Oct 2018) by Marc Salzmann

Comments to the Author:

Thank you very much for submitting a revised manuscript. We received one reviewer report which you can find below. I agree with this report. Further revisions are necessary before your manuscript can be published. Because the overall assessment of your manuscript by the reviewers has been positive, I strongly encourage you to revise your manuscript based on the suggestions in this report.

Dear Editor: Really, really thank you very much for your kindly comments and suggestions. We have followed your suggestions and the reviewer's comments and the corresponding revisions have been made. The changes we made are shown in red font. The revised manuscript with tracked changes is attached later.

Reviewer Report #1:

The authors have responded to most of my minor comments in a satisfactory manner. However, some critical issues prevent the manuscript from being published in its present form, as detailed in the following.

General concern:

The response to my general concern is not clear enough, which also means that the motivation for the analysis of tropopause ascent (rather than tropopause drops) is not outlined well in the manuscript itself. In many of the cases in the manuscript, as well as in Hocker et al 2007, the intrusions are associated with a drop+ascent of the tropopause. I do not fully understand the reference to other atmospheric processes in the response, or how it supports the argument that the intrusions are not related to tropopause drops. Especially the following sentence is unclear: "In other words, various atmospheric processes (and the accompanied tropopause drops) are important conditions for intrusions (or for the strong downdrafts in our study), but intrusion events are not close related to tropopause drops". It is also very confusing to note in the response that the intrusions and the downdrafts are the same thing, while in other places it is mentioned that the downdrafts precede the tropopause ascent and the intrusions. Therefore, a clear distinction of the terms should be made early on in the introduction. In addition, the word "predictor" should be replaced by "diagnostic", for example as it appears in the response "Therefore, we think the strong downdrafts just preceding the rapid tropopause ascent (black bands shown in Fig.13) may serve as a valuable predictor for possible stratospheric intrusions" and in the revised manuscript in line 265 to avoid such confusion. I suggest that the authors revise this part of the text, to convey their motivation better, including English language editing. This issue seems to arise also from the comments of referee #2, and the editor, and deserves careful attention.

**Response:** Really thank the reviewer for the comments. Firstly, we must admit that we cannot conclude that the intrusions are not related to tropopause drops at all. It is from the perspective of quantifying intrusion events that the choice of tropopause ascent (with specific erosion velocity) is more reasonable and feasible than the choice of tropopause drops. This is evidenced by the actual high-resolution observations: (especially with the unique high temporal resolution ozonesonde soundings) by Hocking et al., (2007). The observation results clearly indicate that almost every occurrence of definite stratospheric intrusion is related to a definite RT ascent (>0.2 km/h, occurred at or just before the intrusion). The reverse is also true, that is almost every occurrence of definite RT ascent is associated with some form of intrusions. Therefore, as motivated by this study, tropopause ascent is one of the objects in our manuscript. Please noted that we also cannot conclude that the tropopause ascent is the best and accurate signature that can be used directly for identifying possible intrusions. Therefore, the vertical wind velocity features have also been taken into account in our study. To convey our motivation better, the corresponding sentences in the introduction have been modified, please see the sentences: **'The research by Hocking et al., (2007) have achieved a development in this issue and reported that the rapid ascent in RT altitude (>0.2 km/h) can be a valuable diagnostic for possible stratospheric intrusions. Their observation results clearly indicate that almost every occurrence of definite stratospheric intrusion is related to a definite RT ascent (>0.2 km/h, occurred at or just before the intrusion). The reverse is also true, that is almost**

**every occurrence of definite RT ascent is associated with some form of intrusions. Please noted that we did not mean that the tropopause ascent is the best and most accurate diagnostic that can be used directly for identifying possible intrusions. As motivated by the study of Hocking et al., (2007), tropopause ascent is one of the key objects in this study.'**

Very, very sorry for the confusing response. The vertical motions (downdrafts) and the downward intrusions are not the same thing. To make relationship between the three terms (tropopause ascent, downdrafts, and intrusions) been described clearer, the corresponding sentences in the introduction section have been modified, please see the sentences in the present revised manuscript: **'Using only the information of RT height variability is, of course, insufficient for quantifying intrusion events accurately by radar data. Therefore, radar measurements of vertical motions here are also considered simultaneously to discuss the possible capability of radar measurements for identifying cross-tropopause stratospheric intrusions, which is the main point of this paper. This study is carried out mainly via a detailed case observation during a COL passage and other 20 general cases during various synoptic processes.'**

I'm very sorry about inaccurate use of the word "predictor". In fact, the "predictor" has already been replaced by "diagnostic" in the revised manuscript uploaded in 30 Aug. 2018. Please see the sentence in the present revised manuscript: **'The research by Hocking et al., (2007) have achieved a development in this issue and reported that the rapid ascent in RT altitude (>0.2 km/h) can be a valuable diagnostic for possible stratospheric intrusions'.**

Specific major comments:

2. In the revised manuscript you still state on lines 311-312:"Trajectory results further support the evidence of possible stratospheric intrusions that closely related with the main downdrafts", even though the authors agree that this is not supported by the trajectory analysis.

**Response:**

I'm so sorry. In order to avoid misleading potential readers, the corresponding sentence has been modified to **'Trajectory results further support the evidence of downward intrusions that closely related with the main downdrafts.'**

Specific minor comments:

1. Please enhance and edit the English, as for the general concern above.

Comments 7 and 9: The high-PV at mid levels is not accompanied by high ozone according to Fig. 8. Therefore, it is probable that this is not a major stratospheric intrusion, but rather a diabatically-produced high PV, or a different kind of advection of high PV, but not a clear stratospheric intrusion. This is especially true because the existence of stratospheric intrusions is not supported by the trajectory analysis. This issue is not addressed in the revised manuscript as it should.

**Response:** Really, really thank the reviewer for the valuable comments. According to Figure 8, the high-PV and dry air have been observed intruding deep into troposphere of ~650 hPa. Whereas the vertical structure of AIRS ozone has shown that the enhanced ozone intruded into troposphere of ~500 hPa. From this figure, one thing is clear that stratospheric air (characterized by dry ozone-rich and high-PV) intrusions are indeed occurred and observed (at least deep into 500 hPa). This difference in vertical scale of intrusion between ozone and PV parameters is most likely due to two reasons: 1) the local high PV value observed at ~600 hPa is not a true stratospheric characterized intrusion but rather a diabatically-produced high PV (e.g. Škerlak et al 2015); 2) the relatively poor vertical resolution of AIRS ozone data may have limited the refined observation of the intrusions. In order to make this issue be addressed clearer, the second paragraph of section 3.3 have been modified substantially. Please the sentences in the revised manuscript: **'
[revised manuscript text omitted]

---

## Author Response (AR3)

**Topical Editor Decision: Publish subject to technical corrections** (08 Oct 2018) by Marc Salzmann

Comments to the Author:

Thank you very much for your reply to the reviewer comments and for submitting a revised version of your manuscript.

**Dear Editor:** Really thank you for your help to our paper. I'm also very grateful for the reviewers' valuable comments. Sincerely.

The changes we made are shown in red font. The revised manuscript with tracked changes is attached later.

Suggestions for minor edits:

l. 15: significant -> important l. 18 -> In the light of present understanding, using VHF radars to identify possible stratospheric intrusions still remain unclear -> However, at present several open questions remain regarding the use of VHF radar to identify possible stratospheric intrusions.

**Dear Editor:** We really thank you for pointing out the deficiencies. We have followed the suggestions and modified the corresponding text.

l. 21, l. 22 perhaps use m/s consistently throughout the manuscript (for example also in line 236)

**Dear Editor:** The unit of m/s is used to represent the winds, and km/h is used to denote the tropopause erosion velocity. If we use m/s to represent the tropopause ascent, I'm afraid it will confuse the potential readers, especially those who are familiar of the work by Hocking, (2007). Thus the units are used separately.

l. 31: will gain -> helps to gain l. 38: vertically stable -> stably l. 42: just on the -> the l. 55: are -> is l. 56: but are less, if any -> but is less often, if at all (?)

l. 58 3- -> threel. 60: yield -> obtain a l. 74: seasons -> season l. 92: researches -> research l. 104: The research by Hocking et al., (2007) have achieved a development in this issue and reported -> Hocking et al. (2007) reported l. 105: please define what "RT" means already here (it is only defined further below in line 157)

**Dear Editor:** We really thank you for pointing out these deficiencies. We have followed the suggestions and modified the corresponding text.

l. 109: The reverse is also true ... : here it is not clear whether this statement is still based on Hocking et al. (2007). If it is, it would be good to cite Hocking et al. (2007) once more at the end of this sentence. If not, please explain why this is so.

**Dear Editor:** Indeed, it is based on the statement by Hocking et al. (2007). The citation again is added in the revised manuscript.

l. 110: intrusions -> intrusion
l. 110: noted -> note
l. 120: processes -> situations
l. 137: (Ottersten, 1969) -> Ottersten (1969)
l. 175: is of -> is
l. 180: CH4 -> CH_4 (the "4" should be a subscript)
l. 187: present -> the present
**Dear Editor:** We really thank you for pointing out these deficiencies. We have followed the suggestions and modified the corresponding text.

l. 221: please check how this data should be cited. I think that the usage of this data might be governed by the NCDC use agreement available via https://www.esrl.noaa.gov/psd/data/gridded/data.olrcdr.interp.html.
**Dear Editor:** It is indeed a big problem. We have added the related citation: Lee, (2014); We also have expressed our thanks in the acknowledgment. Please see the sentence: **'The interpolated OLR data provided by the NOAA/OAR/ESRL PSD, Boulder, Colorado, USA, from their Web site at https://www.esrl.noaa.gov/psd/.'**
*Lee, H.-T.: Climate Algorithm Theoretical Basis Document (C-ATBD): Outgoing Longwave Radiation (OLR) - Daily. NOAA's Climate Data Record (CDR) Program, CDRP-ATBD-0526, 46 pp, 2014.*

l. 225: side -> site (also elsewhere)    l. 239: remained -> created (?)
l. 240: where they should be normally high value ->where normally one would expect to find a high value
l. 249: omit "exactly" (I think "well" is enough here and "exactly well" sounds odd to me)
l. 252: cyclone -> cyclones
l. 278: a verb seems to be missing after "is"    l. 306: one thing is -> it is
l. 307: are -> have; observed -> were observed
l. 312: masses parcel -> mass parcel is    l. 317: that -> that are
l. 320: transport -> are transported
l. 322: transportation -> transport
l. 370: Present -> The present
l. 379: majority -> the majority of
l. 409: is served -> served
l. 427: checking -> commenting on (?)
l. 429: thanks -> thank
**Response:** We really thank you for pointing out the deficiencies. We have modified the corresponding text.

**Dear editor:** Personally, I'm sincerely very and very grateful for your help to my paper. All the comments and suggestions are necessary and important.

[revised manuscript text omitted]